# SWAD: Domain Generalization by Seeking Flat Minima

**Junbum Cha**[1][†]    **Sanghyuk Chun**[2][*]    **Kyungjae Lee**[3][*]
**Han-Cheol Cho**[4]    **Seunghyun Park**[4]    **Yunsung Lee**[5]    **Sungrae Park**[6][†]

[1] Kakao Brain    [2] NAVER AI Lab    [3] Chung-Ang University
[4] NAVER Clova    [5] Korea University    [6] Upstage AI Research

## Abstract

Domain generalization (DG) methods aim to achieve generalizability to an unseen target domain by using only training data from the source domains. Although a variety of DG methods have been proposed, a recent study shows that under a fair evaluation protocol, called DomainBed, the simple empirical risk minimization (ERM) approach works comparable to or even outperforms previous methods. Unfortunately, simply solving ERM on a complex, non-convex loss function can easily lead to sub-optimal generalizability by seeking sharp minima. In this paper, we theoretically show that finding flat minima results in a smaller domain generalization gap. We also propose a simple yet effective method, named Stochastic Weight Averaging Densely (SWAD), to find flat minima. SWAD finds flatter minima and suffers less from overfitting than does the vanilla SWA by a dense and overfit-aware stochastic weight sampling strategy. SWAD shows state-of-the-art performances on five DG benchmarks, namely `PACS`, `VLCS`, `OfficeHome`, `TerraIncognita`, and `DomainNet`, with consistent and large margins of +1.6% averagely on out-of-domain accuracy. We also compare SWAD with conventional generalization methods, such as data augmentation and consistency regularization methods, to verify that the remarkable performance improvements are originated from by seeking flat minima, not from better in-domain generalizability. Last but not least, SWAD is readily adaptable to existing DG methods without modification; the combination of SWAD and an existing DG method further improves DG performances. Source code is available at `https://github.com/khanrc/swad`.

## 1 Introduction

Independent and identically distributed (i.i.d.) condition is the underlying assumption of machine learning experiments. However, this assumption may not hold in real-world scenarios, *i.e.*, the training and the test data distribution may differ significantly by *distribution shifts*. For example, a self-driving car should adapt to adverse weather or day-to-night shifts [1, 2]. Even in a simple image recognition scenario, systems rely on wrong cues for their prediction, *e.g.*, geographic distribution [3], demographic statistics [4], texture [5], or backgrounds [6]. Consequently, a practical system should require generalizability to distribution shift, which is yet often failed by traditional approaches.

Domain generalization (DG) aims to address *domain shift* simulated by training and evaluating on different domains. DG tasks assume that both task labels and domain labels are accessible. For example, `PACS` dataset [7] has seven task labels (*e.g.*, "dog", "horse") and four domain labels (*e.g.*, "photo", "sketch"). Previous approaches explicitly reduced domain gaps in the latent space [8–

---

[*]Equal contribution    [†]Part of work done while at NAVER Clova
Correspondence to: Junbum Cha <junbum.cha@kakaobrain.com>, Sungrae Park <sungrae.park@upstage.ai>

35th Conference on Neural Information Processing Systems (NeurIPS 2021).

Table 1: **Comparisons with SOTA.** The proposed SWAD outperforms other state-of-the-art DG methods on five different DG benchmarks with significant gaps (+1.6pp in the average).

|  | PACS | VLCS | OfficeHome | TerraInc | DomainNet | **Avg.** |
|---|---|---|---|---|---|---|
| ERM [29] | 85.5 | 77.5 | 66.5 | 46.1 | 40.9 | 63.3 |
| Best SOTA competitor | 86.6 [30] | 78.8 [31] | 68.7 [31] | 48.6 [32] | 43.6 [15, 33] | 65.3 |
| SWAD (proposed) | **88.1** | **79.1** | **70.6** | **50.0** | **46.5** | **66.9** |
| Previous SOTA [31] + SWAD | 88.3 | 78.9 | 71.3 | 51.0 | 46.8 | 67.3 |

12], obtained well-transferable model parameters by the meta-learning framework [13–16], data augmentation [17–19], or capturing causal relation [20, 21]. Despite numerous previous attempts for a decade, Gulrajani and Lopez-Paz [22] showed that a simple empirical risk minimization (ERM) approach works comparably or even outperforms the previous attempts on diverse DG benchmarks under a fair evaluation protocol, called "DomainBed".

Unfortunately, although ERM showed surprising empirical success on DomainBed, simply minimizing the empirical loss on a complex and non-convex loss landscape is typically not sufficient to arrive at a good generalization [23–26]. In particular, the connection between the generalization gap and the flatness of loss landscapes has been actively discussed under the i.i.d. condition [23–28]. Izmailov et al. [25] argued that seeking flat minima will lead to robustness against the loss landscape shift between training and test datasets, while a simple ERM converges to the boundary of a wide flat minimum and achieves insufficient generalization. In the DG scenario, because training and test loss landscapes differ more drastically due to the domain shift, we conjecture that the generalization gap between flat and sharp minima is larger than expected in the i.i.d. scenario.

To show that flatter minima generalize better to unseen domains, we formulate a robust risk minimization (RRM) problem defined by the worst-case empirical risks within neighborhoods in parameter space [26, 34]. We theoretically show that the generalization gap of DG, *i.e.*, the error on the target domain, is upper bounded by RRM, *i.e.*, a flat optimal solution. Based on our theoretical observation, we modify stochastic weight averaging (SWA) [25], one of the popular existing flatness-aware solvers, by introducing a dense and overfit-aware stochastic weight sampling strategy. First, we suggest to sample weights ***densely***, *i.e.*, for every iteration. Also, we search the start and end iterations for averaging by considering the validation loss to ***avoid overfitting***. We empirically show that the proposed Stochastic Weight Averaging Densely (SWAD) finds flatter minima than the vanilla SWA does, resulting in better generalization to unseen domains.

**Contribution.** Our main contribution is introducing flatness into DG, and showing remarkably outperforming performances against existing DG methods. As shown in Table 1, our SWAD improves the average DG performances by 3.6pp against the ERM baseline and 1.6pp against the existing best methods. Furthermore, by combining SWAD and previous SOTA [31], we even achieve 0.4pp improvements against the vanilla SWAD results. We also empirically show that while popular in-domain generalization methods without considering flatness, *e.g.*, Mixup [35] or CutMix [36], are not effective to out-of-domain generalization (Table 3), flatness-aware methods, *e.g.*, SWA [25] or SAM [26], are only effective methods to both in-domain and out-of-domain generalization.

## 2    A Theoretical Relationship between Flatness and Domain Generalization

Let $\mathcal{D} := \{\mathcal{D}_i\}_i^I$ be a set of training domains, where $\mathcal{D}_i$ is a distribution over input space $\mathcal{X}$, and $I$ is the total number of domains. From each domain, we observe $n$ training data points which consist of input $x$ and target label $y$, $(x_j^i, y_j^i)_{j=1}^n \sim \mathcal{D}_i$. We also define a set of target domain $\mathcal{T} := \{\mathcal{T}_i\}_i^T$ similarly, where the number of target domains $T$ is usually set to one. For the sake of simplicity, unlike Ben-David et al. [37], we assume that there exists a global labeling function $h(x)$ that generates target label for multiple domains, *i.e.*, $y_j^i = h(x_j^i)$ for all $i$ and $j$. Domain generalization (DG) aims to find a model parameter $\theta \in \Theta$ which generalizes well over both multiple training domains $\mathcal{D}$ and unseen target domain $\mathcal{T}$. More specifically, let us consider a bounded instance loss function $\ell : \mathcal{Y} \times \mathcal{Y} \to [0, c]$, such that $\ell(y_1, y_2) = 0$ holds if and only if $y_1 = y_2$ where $\mathcal{Y}$ is a set of labels. For simplicity, we set $c$ to one in our proofs, but we note that $\ell(\cdot, \cdot)$ can be generalized for any bounded loss function. Then, we can define a population loss over multiple domains by $\mathcal{E}_\mathcal{D}(\theta) = \frac{1}{I} \sum_{i=1}^I \mathbb{E}_{x^i \sim \mathcal{D}_i}[\ell(f(x^i; \theta), y^i))]$, where $f(\cdot; \theta)$ is a model parameterized by $\theta$. Formally,

the goal of DG is to find a model which minimizes both $\mathcal{E}_\mathcal{D}(\theta)$ and $\mathcal{E}_\mathcal{T}(\theta)$ by only minimizing an empirical risk $\hat{\mathcal{E}}_\mathcal{D}(\theta) := \frac{1}{In}\sum_{i=1}^{I}\sum_{j=1}^{n}\ell(f(x^i;\theta),y^i))$ over training domains $\mathcal{D}$.

In practice, ERM, *i.e.*, $\arg\min_\theta \hat{\mathcal{E}}_\mathcal{D}(\theta)$, can have multiple solutions that provide similar values of the training losses but significantly different generalizability on $\mathcal{E}_\mathcal{D}(\theta)$ and $\mathcal{E}_\mathcal{T}(\theta)$. Unfortunately, the typical optimization methods, such as SGD and Adam [38], often lead sub-optimal generalizability as finding sharp and narrow minima even under the i.i.d. assumption [23–28]. In the DG scenario, the generalization gap between empirical loss and target domain loss becomes even worse due to domain shift. Here, we provide a theoretical interpretation of the relationship between finding a flat minimum and minimizing the domain generalization gap, inspired by previous studies [23–28].

We consider a robust empirical loss function defined by the worst-case loss within neighborhoods in the parameter space as $\hat{\mathcal{E}}_\mathcal{D}^\gamma(\theta) := \max_{\|\Delta\|\leq\gamma}\hat{\mathcal{E}}_\mathcal{D}(\theta+\Delta)$, where $\|\cdot\|$ denotes the L2 norm and $\gamma$ is a radius which defines neighborhoods of $\theta$. Intuitively, if $\gamma$ is sufficiently larger than the "radius" of a sharp optimum $\theta_s$ of $\hat{\mathcal{E}}_\mathcal{D}(\theta)$, $\theta_s$ is no longer an optimum of $\hat{\mathcal{E}}_\mathcal{D}^\gamma(\theta)$ as well as its neighborhoods within the $\gamma$-ball. On the other hand, if an optimum $\theta_f$ has larger "radius" than $\gamma$, there exists a local optimum within $\gamma$-ball – See Figure 1. Hence, solving the robust risk minimization (RRM), *i.e.*, $\arg\min_\theta \hat{\mathcal{E}}_\mathcal{D}^\gamma(\theta)$, will find

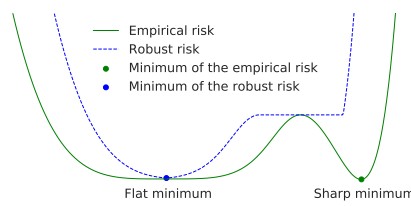

Figure 1: **Robust risk minimization (RRM) and flat minima.** With proper $\gamma$, RRM will find flat minima.

a near solution of a flat optimum showing better generalizability [26, 34]. However, as domain shift worsen the generalization gap by breaking the i.i.d. assumption, it is not trivial that RRM will find an optimum with better DG performance. To answer the question, we first show the generalization bound between $\hat{\mathcal{E}}_\mathcal{D}^\gamma$ and $\mathcal{E}_\mathcal{T}$ as follows:

**Theorem 1.** *Consider a set of $N$ covers $\{\Theta_k\}_{k=1}^{N}$ such that the parameter space $\Theta \subset \cup_k^N \Theta_k$ where $diam(\Theta) := \sup_{\theta,\theta'\in\Theta}\|\theta-\theta'\|_2$, $N := \left\lceil (diam(\Theta)/\gamma)^d \right\rceil$ and $d$ is dimension of $\Theta$. Let $v_k$ be a VC dimension of each $\Theta_k$. Then, for any $\theta \in \Theta$, the following bound holds with probability at least $1-\delta$,*

$$\mathcal{E}_\mathcal{T}(\theta) < \hat{\mathcal{E}}_\mathcal{D}^\gamma(\theta) + \frac{1}{2I}\sum_{i=1}^{I}\mathbf{Div}(\mathcal{D}_i,\mathcal{T}) + \max_{k\in[1,N]}\sqrt{\frac{v_k\ln(m/v_k)+\ln(N/\delta)}{m}}, \tag{1}$$

*where $m = nI$ is the number of the training samples and $\mathbf{Div}(\mathcal{D}_i,\mathcal{T}) := 2\sup_A |\mathbb{P}_{\mathcal{D}_i}(A)-\mathbb{P}_\mathcal{T}(A)|$ is a divergence between two distributions.*

Proof can be done similarly as [37] and [34]. In Theorem 1, the test loss $\mathcal{E}_\mathcal{T}(\theta)$ is bounded by three terms: (1) the robust empirical loss $\hat{\mathcal{E}}_\mathcal{D}^\gamma(\theta)$, (2) the discrepancy between training distribution and test distribution, *i.e.*, the quantity of domain shift, and (3) a confidence bound related to the radius $\gamma$ and the number of the training samples $m$. Our theorem is similar to Ben-David et al. [37], while our theorem does not have the term related to the difference in labeling functions across the domains. It is because we simply assume there is no difference between labeling functions for each domain for simplicity. If one assumes a different labeling function, the dissimilarity term can be derived easily because it is independent and compatible with our main proof. More details of Theorem 1, including proof and discussions on the confidence bound, are in Appendix C.1 and C.2.

From Theorem 1, one can conjure that minimizing the robust empirical loss is directly related to the generalization performances on the target distribution. We show that the domain generalization gap on the target domain $\mathcal{T}$ by the optimal solution of RRM, $\hat{\theta}^\gamma$, is upper bounded as follows:

**Theorem 2.** *Let $\hat{\theta}^\gamma$ denote the optimal solution of the RRM, i.e., $\hat{\theta}^\gamma := \arg\min_\theta \hat{\mathcal{E}}_\mathcal{D}^\gamma(\theta)$, and let $v$ be a VC dimension of the parameter space $\Theta$. Then, the gap between the optimal test loss, $\min_{\theta'}\mathcal{E}_\mathcal{T}(\theta')$, and the test loss of $\hat{\theta}^\gamma$, $\mathcal{E}_\mathcal{T}(\hat{\theta}^\gamma)$, has the following bound with probability at least $1-\delta$.*

$$\begin{aligned}\mathcal{E}_\mathcal{T}(\hat{\theta}^\gamma) - \min_{\theta'}\mathcal{E}_\mathcal{T}(\theta') \quad\leq\quad & \hat{\mathcal{E}}_\mathcal{D}^\gamma(\hat{\theta}^\gamma) - \min_{\theta''}\hat{\mathcal{E}}_\mathcal{D}(\theta'') + \frac{1}{I}\sum_{i=1}^{I}\mathbf{Div}(\mathcal{D}_i,\mathcal{T}) \\ & + \max_{k\in[1,N]}\sqrt{\frac{v_k\ln(m/v_k)+\ln(2N/\delta)}{m}} + \sqrt{\frac{v\ln(m/v)+\ln(2/\delta)}{m}}\end{aligned} \tag{2}$$

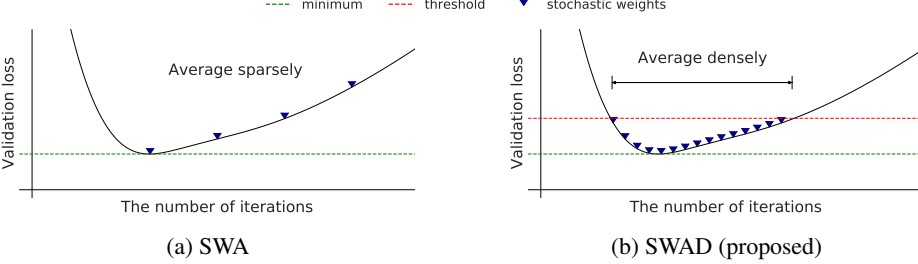

(a) SWA

(b) SWAD (proposed)

Figure 2: **Comparison between SWA and SWAD.** (a) SWA collects stochastic weights for every $K$ epochs from the pre-defined $K_0$ epochs to the final epoch. (b) Our SWAD collects stochastic weights **densely**, *i.e.*, for every iteration, to obtain sufficiently many weights. SWAD collects the weights from the start iteration $t_s$ to the end iteration $t_e$, where $t_s$ and $t_e$ are obtained by monitoring the validation loss (*overfit-aware* scheduling).

Proof is in Appendix C.3. It implies that if we find the optimal solution of the RRM (*i.e.*, $\hat{\theta}^\gamma$), then the generalization gap in the test domain (*i.e.*, $\mathcal{E}_\mathcal{T}(\hat{\theta}^\gamma) - \min_{\theta'} \mathcal{E}_\mathcal{T}(\theta')$) is upper bounded by the gap between the RRM and ERM (*i.e.*, $\hat{\mathcal{E}}_\mathcal{D}^\gamma(\hat{\theta}^\gamma) - \min_{\theta''} \hat{\mathcal{E}}_\mathcal{D}(\theta'')$). Other terms in Theorem 2 are the discrepancy between the train domains $\mathcal{D}$ and the target domain $\mathcal{T}$, and the confidence bounds caused by sample means. We remark that if we choose a proper $\gamma$, the optimal solution of the RRM will find a point near a flat optimum of ERM as shown in Figure 1. Hence, Theorem 2 and the intuition from Figure 1 imply that seeking a flat minimum of ERM will lead to a better domain generalization gap.

## 3 SWAD: Domain Generalization by Seeking Flat Minima

We have shown that flat minima will bring a better domain generalization. In this section, we propose Stochastic Weight Averaging Densely (SWAD) algorithm, and provide empirical quantitative and qualitative analyses on SWAD and flatness to understand why SWAD works better than ERM.

### 3.1 A baseline method: stochastic weight averaging

Since the importance of flatness in loss landscapes has emerged [23–28], several methods have been proposed to find flat minima [25, 26, 39]. We select stochastic weight averaging (SWA) [25] as a baseline, which finds flat minima by a weight ensemble approach. More specifically, SWA updates a pretrained model (namely, a model trained with sufficiently enough training epochs, $K_0$) with a cyclical [40] or high constant learning rate scheduling. SWA gathers model parameters for every $K$ epochs during the update and averages them for the model ensemble. SWA finds an ensembled solution of different local optima found by a sufficiently large learning rate to escape a local minimum. Izmailov et al. [25] empirically showed that SWA finds flatter minima than ERM. We also considered sharpness-aware minimization (SAM) [26], which is another popular flatness-aware solver, but SWA finds flatter minima than SAM (See Figure 3). We illustrate an overview of SWA in Figure 2a.

### 3.2 Dense and overfit-aware stochastic weight sampling strategy

Despite its advantages, directly applying SWA to DG task has two problems. First, SWA averages a few weights (usually less than ten) by sampling weights for every $K$ epochs, results in an inaccurate approximation of flat minima on a high-dimensional parameter space (*e.g.*, 23M for ResNet-50 [41]). Furthermore, a common DG benchmark protocol uses relatively small training epochs (*e.g.*, Gulrajani and Lopez-Paz [22] trained with less than two epochs for `DomainNet` benchmark), resulting in insufficient stochastic weights for SWA. From this motivation, we propose a *"dense"* sampling strategy for gathering sufficiently enough stochastic weights.

In addition, widely used DG datasets, such as `PACS` ($\approx 10K$ images, 7 classes) and `VLCS` ($\approx 11K$ images, 5 classes), are relatively smaller than large-scale datasets, such as ImageNet [42] ($\approx 1.2M$ images, 1K classes). In this case, we observe that a simple ERM approach is rapidly reached to a local optimum only within a few epochs, and easily suffers from the overfitting issue, *i.e.*, the validation loss is increased after a few training epochs. It implies that directly applying the vanilla SWA will suffer from the overfitting issue by averaging sub-optimal solutions (*i.e.*, overfitted parameters). Hence, we need an *"overfit-aware"* sampling scheduling to omit the sub-optimal solutions for SWA.

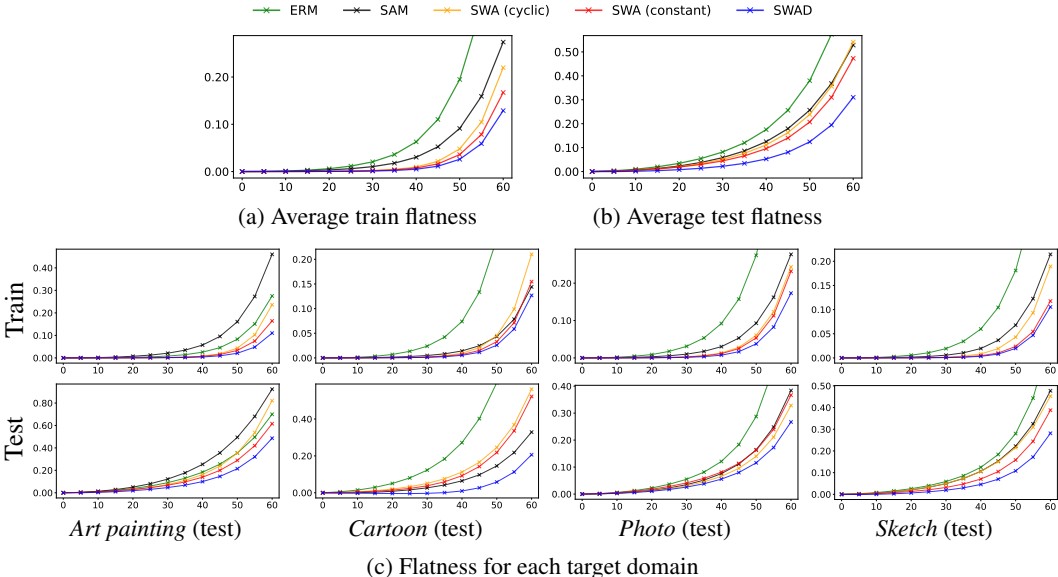

Figure 3: **Local flatness comparisons.** We plot the local flatness via loss gap, *i.e.*, $\mathcal{F}_\gamma(\theta) = \mathbb{E}_{\|\theta'\|=\|\theta\|+\gamma}[\mathcal{E}(\theta') - \mathcal{E}(\theta)]$, of ERM, SAM, SWA, and SWAD by varying radius $\gamma$ on different domains of PACS dataset. For each figure, Y-axis indicates the flatness $\mathcal{F}_\gamma(\theta)$ and X-axis indicates the radius $\gamma$. We measure the train flatness $\mathcal{F}_\gamma^{\mathcal{D}}(\theta)$ on seen domains and the test flatness $\mathcal{F}_\gamma^{\mathcal{T}}(\theta)$ on unseen domain. Each point is computed by Monte-Carlo approximation with 100 random samples. This comparisons show SWAD finds flatter minima than not only ERM but also SAM and SWA.

The main idea of Stochastic Weight Averaging Densely (SWAD) is a dense and overfit-aware stochastic weight gathering strategy. First, instead of collecting weights for every $K$ epochs, SWAD collects weights for every iteration. This dense sampling strategy easily collects sufficiently many weights than the sparse one. We also employ overfit-aware sampling scheduling by considering traces of the validation loss. Instead of sampling weights from $K_0$ pretraining epochs to the final epoch, we search the start iteration (when the validation loss achieves a local optimum for the first time) and the end iteration (when the validation loss is no longer decreased, but keep increasing). More specifically, we introduce three parameters: an optimum patient parameter $N_s$, an overfitting patient parameter $N_e$, and the tolerance rate $r$ for searching the start iteration $t_s$ and the end iteration $t_e$. First, we search $t_s$ which satisfies $\min_{i \in [0,...,N_s-1]} \mathcal{E}_{\text{val}}^{(t_s+i)} = \mathcal{E}_{\text{val}}^{(t_s)}$, where $\mathcal{E}_{\text{val}}^{(i)}$ denotes the validation loss at iteration $i$. Simply, $t_s$ is the first iteration where the loss value is no longer decreased during $N_s$ iterations. Then, we find $t_e$ satisfying $\min_{i \in [0,1,...,N_e-1]} \mathcal{E}_{\text{val}}^{(t_e+i)} > r\mathcal{E}_{\text{val}}^{(t_s)}$. In other words, $t_e$ is the first iteration where the validation loss values exceed the tolerance $r$ during $N_e$ iterations.

We illustrate the overview of SWAD and the comparison of SWAD to SWA in Figure 2. Detailed pseudo code is provided in Appendix B.4. We compare SWAD with other possible SWA strategies in §4.3 and show that our design choice works better for DG tasks.

### 3.3 Empirical analysis of SWAD and flatness

Here, we analyze solutions found by SWAD in terms of flatness. We first verify that the SWAD solution is flatter than those of ERM, SWA, and SAM. Our loss surface visualization shows that the SWAD solution is located on the center of the flat region, while ERM finds a boundary solution. Finally, we show that the sharp boundary solutions by ERM are not generalized well, resulting in sensitivity to the model selection. All following empirical analyses are conducted on PACS dataset, validating by all four domains (art painting, cartoon, photo, and sketch).

**Local flatness anaylsis.** To begin with, we quantify the local flatness of a model parameter $\theta$ by assuming that flat minima will have smaller changes of loss value within its neighborhoods than sharp minima. For the given model parameter $\theta$, we compute the expected loss value changes between $\theta$ and parameters on the sphere surrounding $\theta$ with radius $\gamma$, *i.e.*, $\mathcal{F}_\gamma(\theta) = \mathbb{E}_{\|\theta'\|=\|\theta\|+\gamma}[\mathcal{E}(\theta') - \mathcal{E}(\theta)]$. In

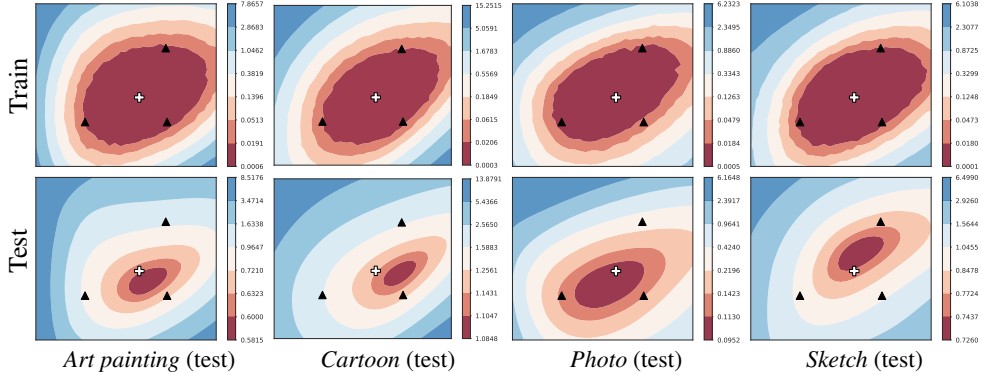

Figure 4: **Loss surfaces on model parameters in `PACS` dataset for each target domain.** The three triangles indicate model weights chosen at the end of training phase with equal intervals. Each plane is defined by the three weights and losses upon the plane are visualized with contours. The center cross mark is averaged point of the three weights. The first and second rows show the averaged training loss and the test loss surfaces, respectively.

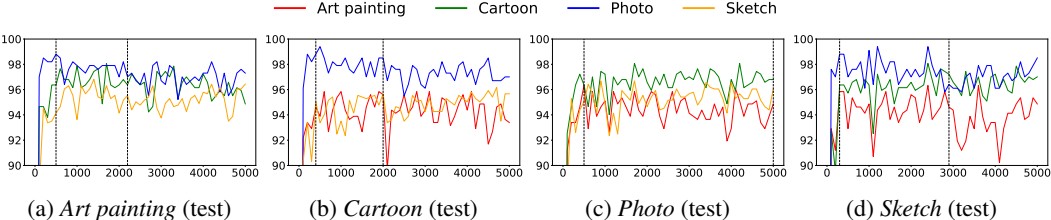

Figure 5: **Validation accuracies for in-domains.** The X- and Y-axis indicate the training iterations and accuracy, respectively, about the validation domains (legend) and the test domain (caption). The vertical dot lines represent start and end iterations, $t_s$ and $t_e$, identified by the overfit-aware sampling strategy of SWAD.

practice, $\mathcal{F}_\gamma(\theta)$ is approximated by Monte-Carlo sampling with 100 samples. Note that the proposed local flatness $\mathcal{F}_\gamma(\theta)$ is computationally efficient than measuring curvature using the Hessian-based quantities. Also, $\mathcal{F}_\gamma(\theta)$ has an unbiased finite sample estimator, while the worst-case loss value, *i.e.*, $\max_{\|\theta'\|=\|\theta\|+\gamma}[\mathcal{E}(\theta') - \mathcal{E}(\theta)]$ has no unbiased finite sample estimator.

In Figure 3, we compare $\mathcal{F}_\gamma(\theta)$ of ERM, SAM, SWA with cyclic learning rate, SWA with constant learning rate, and SWAD by varying radius $\gamma$. SAM and SWA find the solutions with lower local flatness than ERM on average. SWAD finds the most flat minimum in every experiment.

**Loss surface visualization.** We visualize the loss landscapes by choosing three model weights on the optimization trajectory $(\theta_1, \theta_2, \theta_3)$[2], and computing the loss values by linear combinations of $\theta_1, \theta_2, \theta_3$[3] as [25]. More details are in Appendix B.5. In Figure 4, we observe that for all cases, ERM solutions are located at the boundary of a flat minimum of training loss, resulting in poor generalizability in test domains, that is aligned with our theoretical analysis and empirical flatness analysis. Since ERM solutions are located on the boundary of a flat loss surface, we observe that ERM solutions are very sensitive to model selection. In Figure 5, we illustrate the validation accuracies for each train-test domain combination of `PACS` by ERM, over training iterations (one epoch is equivalent to 83 iterations). We first observe that ERM rapidly reaches the best accuracy within only a few training epochs, namely less than 6 epochs. Furthermore, the ERM validation accuracies fluctuate a lot, and the final performance is very sensitive to the model selection criterion.

On the other hand, we observe that SWA solutions are located on the center of the training loss surfaces as well as of the test loss surfaces (Figure 4). Also, our overfit-aware stochastic weight gathering strategy (denoted as the vertical dot lines in Figure 5) prevents the ensembled weight from overfitting and makes SWAD model selection-free.

---

[2]We choose weights at iteration 2500, 3500, 4500 during the training.

[3]Each point is defined by two axes $u$ and $v$ computed by $u = \theta_2 - \theta_1$ and $v = \frac{(\theta_3 - \theta_1) - \langle \theta_3 - \theta_1, \theta_2 - \theta_1 \rangle}{\|\theta_2 - \theta_1\|^2 \cdot (\theta_2 - \theta_1)}$.

Table 2: **Comparison with domain generalization methods and SWAD.** Out-of-domain accuracies on five domain generalization benchmarks are shown. We highlight the **best results** and the second best results. Note that ERM (reproduced), Mixstyle are reproduced numbers, and other numbers are from the original literature and Gulrajani and Lopez-Paz [22] (denoted with †). Our experiments are repeated three times.

| Algorithm | PACS | VLCS | OfficeHome | TerraInc | DomainNet | Avg. |
|---|---|---|---|---|---|---|
| MASF [14] | 82.7 | - | - | - | - | - |
| DMG [33] | 83.4 | - | - | - | 43.6 | - |
| MetaReg [15] | 83.6 | - | - | - | 43.6 | - |
| ER [12] | 85.3 | - | - | - | - | - |
| pAdaIN [47] | 85.4 | - | - | - | - | - |
| EISNet [48] | 85.8 | - | - | - | - | - |
| DSON [30] | 86.6 | - | - | - | - | - |
| ERM† [29] | 85.5 | 77.5 | 66.5 | 46.1 | 40.9 | 63.3 |
| ERM (reproduced) | 84.2 | 77.3 | 67.6 | 47.8 | 44.0 | 64.2 |
| IRM† [20] | 83.5 | 78.6 | 64.3 | 47.6 | 33.9 | 61.6 |
| GroupDRO† [49] | 84.4 | 76.7 | 66.0 | 43.2 | 33.3 | 60.7 |
| I-Mixup† [50–52] | 84.6 | 77.4 | 68.1 | 47.9 | 39.2 | 63.4 |
| MLDG† [13] | 84.9 | 77.2 | 66.8 | 47.8 | 41.2 | 63.6 |
| CORAL† [31] | 86.2 | 78.8 | 68.7 | 47.7 | 41.5 | 64.5 |
| MMD† [53] | 84.7 | 77.5 | 66.4 | 42.2 | 23.4 | 58.8 |
| DANN† [9] | 83.7 | 78.6 | 65.9 | 46.7 | 38.3 | 62.6 |
| CDANN† [10] | 82.6 | 77.5 | 65.7 | 45.8 | 38.3 | 62.0 |
| MTL† [54] | 84.6 | 77.2 | 66.4 | 45.6 | 40.6 | 62.9 |
| SagNet† [32] | 86.3 | 77.8 | 68.1 | 48.6 | 40.3 | 64.2 |
| ARM† [16] | 85.1 | 77.6 | 64.8 | 45.5 | 35.5 | 61.7 |
| VREx† [21] | 84.9 | 78.3 | 66.4 | 46.4 | 33.6 | 61.9 |
| RSC† [55] | 85.2 | 77.1 | 65.5 | 46.6 | 38.9 | 62.7 |
| Mixstyle [17] | 85.2 | 77.9 | 60.4 | 44.0 | 34.0 | 60.3 |
| SWAD (ours) | **88.1** (±0.1) | **79.1** (±0.1) | **70.6** (±0.2) | **50.0** (±0.3) | **46.5** (±0.1) | **66.9** |

# 4 Experiments

## 4.1 Evaluation protocols

**Dataset and optimization protocol.** Following Gulrajani and Lopez-Paz [22], we exhaustively evaluate our method and comparison methods on various benchmarks: PACS [7] (9,991 images, 7 classes, and 4 domains), VLCS [43] (10,729 images, 5 classes, and 4 domains), OfficeHome [44] (15,588 images, 65 classes, and 4 domains), TerraIncognita [45] (24,788 images, 10 classes, and 4 domains), and DomainNet [46] (586,575 images, 345 classes, and 6 domains).

For a fair comparison, we follow training and evaluation protocol by Gulrajani and Lopez-Paz [22], including the dataset splits, hyperparameter (HP) search and model selection (while SWAD does not need it) on the validation set, and optimizer HP, except the HP search space and the number of iterations for DomainNet. We use a reduced HP search space to reduce the computational costs. We also tripled the number of iterations for DomainNet from 5,000 to 15,000 because we observe that 5,000 is not sufficient to convergence. We re-evaluate ERM with 15,000 iterations, and observe 3.1pp average performance improvement ($40.9\% \rightarrow 44.0\%$) in DomainNet. For training, we choose a domain as the target domain and use the remaining domains as the training domain where 20% samples are used for validation and model selection. ImageNet [42] trained ResNet-50 [41] is employed as the initial weight, and optimized by Adam [38] optimizer with a learning rate of 5e-5. We construct a mini-batch containing all domains where each domain has 32 images. We set SWAD HPs $N_s$ to 3, $N_e$ to 6, and $r$ to 1.2 for VLCS and 1.3 for the others by HP search on the validation sets. Additional implementation details, such as other HPs, are given in Appendix B.

**Evaluation metrics.** We report out-of-domain accuracies for each domain and their average, *i.e.*, a model is trained and validated on training domains and evaluated on the unseen target domain. Each out-of-domain performance is an average of three different runs with different train-validation splits.

## 4.2 Main results

**Comparison with domain generalization methods.**
We report the full out-of-domain performances on five DG benchmarks in Table 2. The full tables including out-of-domain accuracies for each domain are in Appendix E. In all experiments, our SWAD achieves significant performance gain against ERM as well as the previous best results: +2.6pp in `PACS`, +0.3pp in `VLCS`, +1.4pp in `TerraIncognita`, +1.9pp in `OfficeHome`, and +2.9pp in `DomainNet` comparing to the previous best results. We observe that SWAD provides two practical advantages comparing to previous methods. First, SWAD does not need any modification on training objectives or model architecture, *i.e.*, it is universally applicable to any other methods. As an example, we show that SWAD actually improves the performances of other DG methods, such as CORAL [31] in Table 4. Moreover, as we discussed before, SWAD is free to the model selection,

Table 3: **Comparison between generalization methods on** `PACS`**.** The scores are averaged over all settings using different target domains. (↑) and (↓) indicate statistically significant improvement and degradation from ERM.

|  | Out-of-domain | In-domain |
| --- | --- | --- |
| ERM | $85.3_{\pm 0.4}$ | $96.6_{\pm 0.0}$ |
| EMA | $85.5_{\pm 0.4}$(-) | $97.0_{\pm 0.1}$(↑) |
| SAM | $85.5_{\pm 0.1}$(-) | $97.4_{\pm 0.1}$(↑) |
| Mixup | $84.8_{\pm 0.3}$(-) | $97.3_{\pm 0.1}$(↑) |
| CutMix | $83.8_{\pm 0.4}$(↓) | $97.6_{\pm 0.1}$(↑) |
| VAT | $85.4_{\pm 0.6}$(-) | $96.9_{\pm 0.2}$(↑) |
| Π-model | $83.5_{\pm 0.5}$(↓) | $96.8_{\pm 0.2}$(↑) |
| SWA | $85.9_{\pm 0.1}$(↑) | $97.1_{\pm 0.1}$(↑) |
| SWAD | $\mathbf{87.1}_{\pm 0.2}$(↑) | $\mathbf{97.7}_{\pm 0.1}$(↑) |

resulting in stable performances (*i.e.*, small standard errors) on various benchmarks. Note that we only compare results with ResNet-50 backbone for a fair comparison. We describe the implementation details of each comparison method and the hyperparameter search protocol in Appendix B.

**Comparison with conventional generalization methods.** We also compare SWAD with other conventional generalization methods to show that the remarkable domain generalization gaps by SWAD is not achieved by better generalization, but by seeking flat minima. The comparison methods include flatness-aware optimization methods, such as SAM [26], ensemble methods, such as EMA [56], data augmentation methods, such as Mixup [35] and CutMix [36], and consistency regularization methods, such as VAT [57] and Π-model [58]. We also split in-domain datasets into training (60%), validation (20%), and test (20%) splits, while no in-domain test set used for Table 2. Every experiment is repeated three times.

The results are shown in Table 3. We observe that all conventional methods helps in-domain generalization, *i.e.*, performing better than ERM on in-domain test set. However, their out-of-domain performances are similar to or even worse than ERM. For example, CutMix and Π-model improve in-domain performances by 1.0pp and 0.2pp but degrade out-of-domain performances by 1.5pp and 1.8pp. SAM, another method for seeking flat minima, slightly increases both in-domain and out-of-domain performances but the out-of-domain performance is not statistically significant. We will discuss performances of SAM in other benchmarks later. In contrast, the vanilla SWA and our SWAD significantly improve both in-domain and out-of-domain performances. SWAD improves the performances by SWA with statistically significantly gaps: 1.2pp on the out-of-domain and 0.6pp on the in-domain. Further comparison between SWA and SWAD is provided in §4.3.

Table 4: **Combination of SWAD and other methods.** The scores are averaged over every target domain case. The performances of ERM, CORAL, and SAM are optimized by HP searches of DomainBed. In contrast, for the SWAD combination cases, CORAL and SAM use default HPs without additional HP search. We additionally compare SWAD to SWA$_{\text{w/ const}}$. Note that ERM + SWAD is same as "SWAD" in Table 2.

|  | PACS | VLCS | OfficeHome | TerraInc | DomainNet | Avg. (Δ) |
| --- | --- | --- | --- | --- | --- | --- |
| ERM | $85.5_{\pm 0.2}$ | $77.5_{\pm 0.4}$ | $66.5_{\pm 0.3}$ | $46.1_{\pm 1.8}$ | $40.9_{\pm 0.1}$ | 63.3 |
| ERM + SWA$_{\text{w/ const}}$ | $86.9_{\pm 0.2}$ | $76.6_{\pm 0.1}$ | $69.3_{\pm 0.3}$ | $49.2_{\pm 1.2}$ | $45.9_{\pm 0.0}$ | 65.6 (+2.3) |
| ERM + SWAD | $88.1_{\pm 0.1}$ | $79.1_{\pm 0.1}$ | $70.6_{\pm 0.2}$ | $50.0_{\pm 0.3}$ | $46.5_{\pm 0.1}$ | 66.9 (+3.6) |
| CORAL | $86.2_{\pm 0.3}$ | $78.8_{\pm 0.6}$ | $68.7_{\pm 0.3}$ | $47.6_{\pm 1.0}$ | $41.5_{\pm 0.1}$ | 64.5 |
| CORAL + SWAD | $88.3_{\pm 0.1}$ | $78.9_{\pm 0.1}$ | $71.3_{\pm 0.1}$ | $51.0_{\pm 0.1}$ | $46.8_{\pm 0.0}$ | 67.3 (+2.8) |
| SAM | $85.8_{\pm 0.2}$ | $79.4_{\pm 0.1}$ | $69.6_{\pm 0.1}$ | $43.3_{\pm 0.7}$ | $44.3_{\pm 0.0}$ | 64.5 |
| SAM + SWAD | $87.1_{\pm 0.2}$ | $78.5_{\pm 0.2}$ | $69.9_{\pm 0.1}$ | $45.3_{\pm 0.9}$ | $46.5_{\pm 0.1}$ | 65.5 (+1.0) |

**Combinations with other methods.** Since SWAD does not require any modification on training procedures and model architectures, SWAD is universally applicable to any other methods. Here, we combine SWAD with ERM, CORAL [31], and SAM [26]. Results are shown in Table 4. Both CORAL and SAM solely show better performances than ERM with +1.2pp average out-of-domain accuracy gap. Note that SAM is not a DG method but a sharpness-aware optimization method to find flat minima. It supports our theoretical motivation: DG can be achieved by seeking flat minima.

By applying SWAD on the baselines, the performances are consistently improved by 3.6pp on ERM, 2.8pp on CORAL, and 1.0pp on SAM. Interestingly, CORAL + SWAD show the best performances with both incorporating different advantages of utilizing domain labels and seeking flat minima. We also observe that SAM + SWAD shows worse performance than ERM + SWAD, while SAM performs better than ERM. We conjecture that it is because the objective control by SAM restricts the model parameter diversity durinig training, reducing the diversity for SWA ensemble. However, applying SWAD on SAM still leads to better performances than the sole SAM. The results demonstrate that the application of SWAD on other baselines is a simple yet effective method for DG.

## 4.3 Ablation study

Table 5: **Ablation studies of the stochastic weights selection strategies on PACS and VLCS.** In the configuration, "$t_s$", "$t_e$", "lr", and "interval" indicate start and end iterations of sampling, a learning rate schedule, and a stochastic weight sampling interval, respectively. "Opt" and "Overfit" indicate the start and end iterations identified by our overfit-aware sampling strategy, and "Val" means the start and end iterations whose averaging shows the best accuracy on the validation set. "Cyclic" and "Const" represent cyclic and constant learning rate schedules. All experiments are repeated three times.

| | Configuration | | | | Out-of-domain | | | In-domain | | |
| | $t_s$ | $t_e$ | lr | interval | PACS | VLCS | Avg. | PACS | VLCS | Avg. |
|---|---|---|---|---|---|---|---|---|---|---|
| SWA$_{\text{w/ cyclic}}$ | 4000 | 5000 | Cyclic | 100 | 85.9 ±0.1 | 76.6 ±0.1 | 81.2 | 97.1 ±0.1 | 85.0 ±0.2 | 91.0 |
| SWA$_{\text{w/ const}}$ | 4000 | 5000 | Const | 100 | 86.5 ±0.3 | 76.7 ±0.2 | 81.6 | 97.3 ±0.1 | 85.0 ±0.2 | 91.1 |
| SWAD$_{\text{w/o Dense}}$ | Opt | Overfit | Const | 100 | 86.5 ±0.4 | 78.0 ±0.7 | 82.2 | 97.6 ±0.1 | 85.8 ±0.4 | 91.7 |
| SWAD$_{\text{w/o Opt-Overfit}}$ | 4000 | 5000 | Const | 1 | 86.6 ±0.6 | 76.9 ±0.3 | 81.7 | 97.5 ±0.1 | 85.2 ±0.1 | 91.3 |
| SWAD$_{\text{w/o Overfit}}$ | Opt | 5000 | Const | 1 | **87.1** ±0.3 | 77.6 ±0.1 | 82.4 | **97.7** ±0.1 | 85.8 ±0.3 | 91.8 |
| SWAD$_{\text{fit-on-val}}$ | Val | Val | Const | 1 | 86.2 ±0.2 | 78.6 ±0.1 | 82.4 | 97.5 ±0.2 | 85.8 ±0.3 | 91.7 |
| SWAD (proposed) | Opt | Overfit | Const | 1 | **87.1** ±0.2 | **78.9** ±0.2 | **83.0** | **97.7** ±0.1 | **86.1** ±0.5 | **91.9** |

Table 5 provides ablative studies on the starting and ending iterations for averaging, the learning rate schedule, and the sampling interval. SWA$_{\text{w/ cyclic}}$ (SWA in Table 3) and SWA$_{\text{w/ constant}}$ are vanilla SWAs with fixed sampling positions. We also report SWAD by eliminating three factors: the dense sampling strategy, and searching the start iteration, searching the end iteration. The dense sampling strategy lets SWAD estimate a more accurate approximation of flat minima: showing 0.8pp degeneration in the average out-of-domain accuracy (SWAD$_{\text{w/o Dense}}$). When we take an average from $t_s$ to the final iteration, the out-of-domain performance degrades by 0.6pp (SWAD$_{\text{w/o Overfit}}$). Similarly, a fixed scheduling without the overfit-aware scheduling only shows very marginal improvements from the vanilla SWA (SWAD$_{\text{w/o Opt-Overfit}}$). We also evaluate SWAD$_{\text{fit-on-val}}$ that uses the range achieving the best performances on the validation set, but it becomes overfitted to the validation, results in lower performances than SWAD. The results demonstrate the benefits of combining "dense" and "overfit-aware" sampling strategies of SWAD.

## 4.4 Exploring the other applications: ImageNet robustness

Table 6: **ImageNet robustness benchmarks.** We show the ImageNet generalization performances on ImageNet-C, background challenge (BGC), and ImageNet-R.

| Method | ImageNet (%) ↑ | ImageNet-C (mCE) ↓ | BGC (%) ↑ | ImageNet-R (%) ↑ |
|---|---|---|---|---|
| ERM | 76.5 | 57.6 | 8.7 | 36.7 |
| SWA | 76.9 | 56.8 | 10.9 | 37.5 |
| SWAD (ours) | **77.0** | **55.7** | **11.8** | **38.8** |

Since SWAD does not rely on domain labels, it can be applied to other robustness tasks not containing domain labels. Table 6 show the generalizability of SWAD on ImageNet [42] and its shifted benchmarks, namely, ImageNet-C [59], ImageNet-R [60], and background challenge (BGC) [61]. SWAD consistently improves robustness performances against the ERM baseline and the SWA baseline. These results support that our method is robustly and widely applicable to improve both in-domain and out-of-domain generalizability. The detailed setup is provided in Appendix B.6.

## 5 Discussion and Limitations

Despite many benefits from SWAD, such as the significant performance improvements, model selection-free property, working plug-and-play manner for various methods, there are some potential limitations. Here, we discuss the limitations of SWAD for further improvements.

**Confidence error in Theorem 1.** While the confidence error in Theorem 1 tells the effect of $\gamma$ on generalization error bound, there exists a limitation in that the confidence error term shows improper behavior with respect to $\gamma$ if $\gamma$ is close to zero. The behavior we expect is that the confidence error of RRM converges to the confidence error of ERM as $\gamma$ decreases to zero, however, the current theorem does not show such tendency since the confidence bound diverges to infinity when $\gamma$ goes to zero. However, we would like to note that this limitation is not a drawback of RRM, but it is caused by the looseness of the union bound which is a mathematical technique used to derive the confidence error of RRM. Our RRM formulation has a similarity to previous works [26, 34] and we note that the counter-intuitive behavior of the confidence bound and $\gamma$ also appears in Foret et al. [26].

**SWAD is not a perfect flatness-aware optimization method.** Note that SWAD is not a perfect and theoretically guaranteed solver for flat minima, but a heuristic approximation with empirical benefits. However, even if a better flatness-aware optimization method is proposed, our theoretical contribution still holds: showing the relationship between flat minima and DG.

**SWAD does not strongly utilize domain-specific information.** In Theorem 2, the domain generalization gap is bounded by three factors: flat minima, domain discrepancy, and confidence bound. Most of the existing approaches focus on domain discrepancy, reducing the difference between the source domains and the target domain by domain invariant learning [8–12]. SWAD focuses on the first factor, the flat minima. While the domain labels are used to construct a mini-batch, SWAD does not strongly utilize domain-specific information. It implies that if one can consider both flatness and domain discrepancy, better domain generalization can be achievable. Table 4 gives us a clue: the combination of CORAL (utilizing domain-specific information) and SWAD (seeking flat minima) shows the best performance among all comparison methods. As a future research direction, we encourage studying a method that can achieve both flat optima and small domain discrepancy.

## 6 Concluding Remarks

In this paper, we theoretically and empirically demonstrate that domain generalization (DG) is achievable by seeking flat minima. We propose SWAD that captures flatter minima than the vanilla SWA does. The extensive experiments on five DG benchmarks show superior performances of SWAD compared with existing DG methods. In addition, combinations of SWAD and existing DG methods even show better performances than the vanilla SWAD. We theoretically and empirically observe that seeking flat minima can achieve better generalizability to both in-domain and out-of-domain, while strong in-domain generalization methods without consideration of flatness, *e.g.*, Mixup or CutMix, cannot guarantee to achieve out-of-domain generalizability in both theory and practice. This study first brings the concept of flatness into DG tasks, and shows strong empirical performances not only in DG but also in ImageNet benchmarks. We hope that this study promotes a new research direction of seeking flat minima for domain generalization and other robustness tasks.

## Acknowledgments and Disclosure of Funding

NAVER Smart Machine Learning (NSML) [62] and Kakao Brain Cloud platform have been used in experiments. This work was supported by IITP grant funded by the Korea government (MSIT) (No. 2021-0-01341, AI Graduate School Program, CAU).

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
