# SWAD: Domain Generalization
# by Seeking Flat Minima — Appendix

**Junbum Cha**[1][†]    **Sanghyuk Chun**[2][*]    **Kyungjae Lee**[3][*]
**Han-Cheol Cho**[4]    **Seunghyun Park**[4]    **Yunsung Lee**[5]    **Sungrae Park**[6][†]

[1] Kakao Brain    [2] NAVER AI Lab    [3] Chung-Ang University
[4] NAVER Clova    [5] Korea University    [6] Upstage AI Research

## A  Potential Societal Impacts

In this study, we theoretically and empirically demonstrate that domain generalization (DG) is achievable by seeking flat minima, and propose SWAD to find flat minima. With SWAD, researchers and developers can make a model robust to domain shift in a real deployment environment, without relying on a task-dependent prior, a modified objective function, or a specific model architecture. Accordingly, SWAD has potential positive impacts by developing machines less biased towards ethical aspects, as well as potential negative impacts, *e.g.*, improving weapon or surveillance systems under unexpected environment changes.

## B  Implementation Details

### B.1  Hyperparameters of SWAD

The evaluation protocol by Gulrajani and Lopez-Paz [1] is computationally too expensive; it requires about 4,142 models for every DG algorithm. Hence, we reduce the search space of SWAD for computational efficiency; batch size and learning rate are set to 32 for each domain and 5e-5, respectively. We set dropout probability and weight decay to zero. We only search $N_s$, $N_e$ and $r$. $N_s$ and $N_e$ are searched in `PACS` dataset, and the searched values are used for all experiments, while $r$ is searched in [1.2, 1.3] depending on dataset. As a result, we use $N_s = 3$, $N_e = 6$, and $r = 1.2$ for `VLCS` and $r = 1.3$ for the others. We initialize our model by ImageNet-pretrained ResNet-50 and batch normalization statistics are frozen during training. The number of total iterations is $15,000$ for DomainNet and $5,000$ for others, which are sufficient numbers to be converged. Finally, we slightly modify the evaluation frequency because it should be set to small enough to detect the moments that the model is optimized and overfitted. However, too small frequency brings large evaluation overhead, thus we compromise between exactness and efficiency: 50 for `VLCS`, 500 for `DomainNet`, and 100 for others.

### B.2  Hyperparameter search protocol for reproduced results

We evaluate recently proposed methods, SAM [2] and Mixstyle [3], and compare them with previous results. For a fair comparison, we follow the hyperparameter (HP) search protocol proposed by Gulrajani and Lopez-Paz [1], with a modification to reduce computational resources. They searched HP by training a total of 58,000 models, corresponding to about 4,142 runs for each algorithm. It is too much computational burden to train 4,142 models whenever evaluate a new algorithm. Therefore, we re-design the HP search protocol efficiently and effectively. In the HP search protocol of DomainBed [1], training domains and algorithm-specific parameters are included in the HP search space,

---

[*]Equal contribution    [†]Part of work done while at NAVER Clova
Correspondence to: Junbum Cha <junbum.cha@kakaobrain.com>, Sungrae Park <sungrae.park@upstage.ai>

35th Conference on Neural Information Processing Systems (NeurIPS 2021).

Table 1: **Hyperparameter search space comparison.** U and list indicate Uniform distribution and random choice, respectively.

| Parameter | Default value | DomainBed | Ours |
|---|---|---|---|
| batch size | 32 | $2^{U(3,5.5)}$ | 32 |
| learning rate | 5e-5 | $10^{U(-5,-3.5)}$ | [1e-5, 3e-5, 5e-5] |
| ResNet dropout | 0 | [0.0, 0.1, 0.5] | [0.0, 0.1, 0.5] |
| weight decay | 0 | $10^{U(-6,-2)}$ | [1e-4, 1e-6] |

and HP is found for every data split independently by random search. Instead, we do not sample training domains, use HP found in the first data split to the other splits, search algorithm-specific HP independently, and conduct grid search on the more effectively designed HP space as shown in Table 1. Through the proposed protocol, we find HP for an algorithm under only 396 runs. Although the number of total runs is reduced to about 10% ($4,142 \rightarrow 396$), the results of reproduced ERM is improved 0.9pp in average ($63.3\% \rightarrow 64.2\%$). It demonstrates both the effectiveness and the efficiency of our search protocol.

### B.3  Algorithm-specific hyperparameters

We search the algorithm-specific hyperparameters independently in PACS dataset, based on the values suggested from each paper. For Mixstyle [3], we insert Mixstyle block with domain label after the 1st, 2nd, and 3rd residual blocks with $\alpha = 0.1$ and $p = 0.5$. We train SAM [2] with $\rho = 0.05$, and VAT [4] with $\epsilon = 1.0$ and $\alpha = 1.0$. In Π-model [5], $w_{max} = 1$ is chosen among various $w_{max}$ values such as 1, 10, 100, and 300. We use EMA [6] with $decay = 0.99$, Mixup [7] with $\alpha = 0.2$, and CutMix [8] with $\alpha = 1.0$ and $p = 0.5$.

### B.4  Pseudo code

---
**Algorithm 1:** Stochastic Weight Averaging Densely
---
**Input:** initial weight $\theta_0$, constant learning rate $\alpha$, tolerance rate $r$, optimum patience $N_s$, overfit
       patience $N_e$, total number of iterations $T$, training loss $\mathcal{E}_{\text{train}}^{(i)}$, validation loss $\mathcal{E}_{\text{val}}^{(i)}$

**Output:** averaged weight $\theta^{\text{SWAD}}$ from $t_s$ to $t_e$

1   $t_s \leftarrow 0$              `// start iteration for averaging`
2   $t_e \leftarrow T$              `// end iteration for averaging`
3   $l \leftarrow \text{None}$              `// loss threshold`
4   **for** $i \leftarrow 1$ **to** $T$ **do**
5      $\theta_i \leftarrow \theta_{i-1} - \alpha \nabla \mathcal{E}_{\text{train}}^{(i-1)}$
6      **if** $l = \text{None}$ **then**
7          **if** $\mathcal{E}_{val}^{(i-N_s+1)} = \min_{0 \leq i' < N_s} \mathcal{E}_{val}^{(i-i')}$ **then**
8             $t_s \leftarrow i - N_s + 1$
9             $l \leftarrow \frac{r}{N_s} \sum_{i'=0}^{N_s-1} \mathcal{E}_{\text{val}}^{(i-i')}$
10      **else if** $l < \min_{0 \leq i' < N_e} \mathcal{E}_{val}^{(i-i')}$ **then**
11          $t_e \leftarrow i - N_e$
12          **break**
13   $\theta^{\text{SWAD}} \leftarrow \frac{1}{t_e - t_s + 1} \sum_{i'=t_s}^{t_e} \theta^{i'}$

---

### B.5  Loss surface visualization

Following Garipov et al. [9], we choose three model weights $\theta_1, \theta_2, \theta_3$ and define two dimensional weight plane from the weights:

$$u = \theta_2 - \theta_1, \qquad v = \frac{(\theta_3 - \theta_1) - \langle \theta_3 - \theta_1, \theta_2 - \theta_1 \rangle}{\|\theta_2 - \theta_1\|^2 \cdot (\theta_2 - \theta_1)}, \tag{1}$$

where $\hat{u} = u/\|u\|$ and $\hat{v} = v/\|v\|$ are orthonormal bases of the weight plane. Then, we build Cartesian grid near the weights on the plane. For each grid point, we calculate the weight corresponding to the point and compute loss from the weight. The results are visualized as a contour plot, as shown in Figure 4 in the main text.

### B.6 ImageNet robustness experiments

We investigate the extensibility of SWAD via three robustness benchmarks (Section 4.4 in the main text), namely ImageNet-C [10], ImageNet-R [11], and background challenge (BGC) [12]. ImageNet-C measures the robustness against common corruptions such as Gaussian noise, blur, or weather changes. We follow Hendrycks and Dietterich [10] for measuring mean corruption error (mCE). The lower ImageNet-C implies that the model is robust against corruption noises. BGC evaluates the robustness against background manipulations as well as the adversarial robustness. The BGC dataset has two groups, foreground and background. BGC manipulates images by combining the foregrounds and backgrounds, and measures whether the model predicts a consistent prediction with any manipulated image. ImageNet-R tests the robustness against different domains. ImageNet-R collects very different domain images of ImageNet, such as art, cartoons, deviantart, graffiti, embroidery, graphics, origami, paintings, patterns, plastic objects, plush objects, sculptures, sketches, tattoos, toys, and video game renditions. Showing better performances in ImageNet-R leads to the same conclusion as other domain generalization benchmarks.

**Experiment details.** We use ResNet-50 architecture and mostly follow standard training recipes. We use SGD optimizer with momentum of 0.9, base learning rate of 0.1 with linear scaling rule [13] and polynomial decay, 5 epochs gradual warmup, batch size of 2048, and total epochs of 90. For SWA, the learning rate is decayed to 1/20 until 80% of training (72 epochs), and the cyclic learning rate with 3 epochs cycle length is used for the left 20% of training. SWAD follows the same learning rate decay until 80% of training, but averages every weight from every iteration after 80% of training with constant learning rate.

## C Proof of Theorems

### C.1 Technical Lemmas

Consider an instance loss function $\ell(y_1, y_2)$ such that $\ell : \mathcal{Y} \times \mathcal{Y} \to [0, 1]$ and $\ell(y_1, y_2) = 0$ if and only if $y_1 = y_2$. Then, we can define a functional error as $\mathcal{E}_{\mathcal{P}}(f(\cdot; \theta), h) := \mathbb{E}_{\mathcal{P}}[\ell(f(x; \theta), h(x))]$. Note that if we set $h$ as a true label function which generates the label of inputs, $y = h(x)$, then, it becomes a population loss $\mathcal{E}_{\mathcal{P}}(\theta) = \mathcal{E}_{\mathcal{P}}(f(\cdot; \theta), h)$. Given two distributions, $\mathcal{P}$ and $\mathcal{Q}$, the following lemma shows that the difference between the error with $\mathcal{P}$ and the error with $\mathcal{Q}$ is bounded by the divergence between $\mathcal{P}$ and $\mathcal{Q}$.

**Lemma 1.** $|\mathcal{E}_{\mathcal{P}}(h_1, h_2) - \mathcal{E}_{\mathcal{Q}}(h_1, h_2)| \leq \frac{1}{2}\mathbf{Div}(\mathcal{P}, \mathcal{Q})$

*Proof.* We employ the same technique in Zhao et al. [14] for our loss function $\ell$. From the Fubini's theorem, we have,

$$\mathbb{E}_{x \sim \mathcal{P}}[\ell(h_1(x), h_2(x))] = \int_0^\infty \mathbb{P}_{\mathcal{P}}\left(\ell(h_1(x), h_2(x)) > t\right) dt \tag{2}$$

By using this fact,

$$|\mathbb{E}_{x \sim \mathcal{P}}[\ell(h_1(x), h_2(x))] - \mathbb{E}_{x \sim \mathcal{Q}}[\ell(h_1(x), h_2(x))]| \tag{3}$$

$$= \left| \int_0^\infty \mathbb{P}_\mathcal{P}\left(\ell(h_1(x), h_2(x)) > t\right) dt - \int_0^\infty \mathbb{P}_\mathcal{Q}\left(\ell(h_1(x), h_2(x)) > t\right) dt \right| \tag{4}$$

$$\leq \int_0^\infty \left|\mathbb{P}_\mathcal{P}\left(\ell(h_1(x), h_2(x)) > t\right) - \mathbb{P}_\mathcal{Q}\left(\ell(h_1(x), h_2(x)) > t\right)\right| dt \tag{5}$$

$$\leq M \sup_{t \in [0,M]} \left|\mathbb{P}_\mathcal{P}\left(\ell(h_1(x), h_2(x)) > t\right) - \mathbb{P}_\mathcal{Q}\left(\ell(h_1(x), h_2(x)) > t\right)\right| \tag{6}$$

$$\leq M \sup_{h_1, h_2} \sup_{t \in [0,M]} \left|\mathbb{P}_\mathcal{P}\left(\ell(h_1(x), h_2(x)) > t\right) - \mathbb{P}_\mathcal{Q}\left(\ell(h_1(x), h_2(x)) > t\right)\right| \tag{7}$$

$$\leq M \sup_{\bar{h} \in \bar{\mathcal{H}}} \left|\mathbb{P}_\mathcal{P}\left(\bar{h}(x) = 1\right) - \mathbb{P}_\mathcal{Q}\left(\bar{h}(x) = 1\right)\right| \tag{8}$$

$$\leq M \sup_A \left|\mathbb{P}_\mathcal{P}\left(A\right) - \mathbb{P}_\mathcal{Q}\left(A\right)\right| \tag{9}$$

where $\bar{\mathcal{H}} := \{\mathbb{I}[\ell(h(x), h'(x)) > t] | h, h' \in \mathcal{H}, t \in [0, M]\}$. $\qquad\square$

**Lemma 2.** *Consider a distribution $\mathcal{S}$ on input space and global label function $f : \mathcal{X} \to \mathcal{Y}$. Let $\left\{\Theta_k \subset \mathbb{R}^d, k = 1, \cdots, N\right\}$ be a finite cover of a parameter space $\Theta$ which consists of closed balls with radius $\gamma/2$ where $N := \left\lceil (diam(\Theta)/\gamma)^d \right\rceil$. Let $\theta_k \in \arg\max_{\Theta_k \cap \Theta} \mathcal{E}_\mathcal{S}(\theta)$ be a local maximum in the k-th ball. Let a VC dimension of $\Theta_k$ be $v_k$. Then, for any $\theta \in \Theta$, the following bound holds with probability at least $1 - \delta$.*

$$\mathcal{E}_\mathcal{S}(\theta) - \hat{\mathcal{E}}_\mathcal{S}^\gamma(\theta) \leq \max_k \sqrt{\frac{(v_k\left[\ln\left(n/v_k\right) + 1\right] + \ln\left(N/\delta\right))}{2n}} \tag{10}$$

*where $\hat{\mathcal{E}}_\mathcal{S}^\gamma(\theta_k)$ is an empirical robust risk with $n$ samples.*

*Proof.* We first show that the following inequality holds for the local maximum of $N$ covers,

$$\mathbb{P}\left(\max_k \left[\mathcal{E}_\mathcal{S}(\theta_k) - \hat{\mathcal{E}}_\mathcal{S}(\theta_k)\right] > \epsilon\right) \leq \sum_{k=1}^N \mathbb{P}\left(\mathcal{E}_\mathcal{S}(\theta_k) - \hat{\mathcal{E}}_\mathcal{S}(\theta_k) > \epsilon\right) \tag{11}$$

$$\leq \sum_{k=1}^N \mathbb{P}\left(\sup_{\theta \in \Theta_k}\left[\mathcal{E}_\mathcal{S}(\theta) - \hat{\mathcal{E}}_\mathcal{S}(\theta)\right] > \epsilon\right) \tag{12}$$

$$\leq \sum_{k=1}^N \left(\frac{en}{v_k}\right)^{v_k} e^{-2n\epsilon^2}. \tag{13}$$

Now, we introduce a confidence error bound $\epsilon_k := \sqrt{\frac{(v_k[\ln(n/v_k)+1]+\ln(N/\delta))}{2n}}$. Then, we set $\epsilon := \max_k \epsilon_k$. Then, we get,

$$\mathbb{P}\left(\max_k\left[\mathcal{E}_\mathcal{S}(\theta_k) - \hat{\mathcal{L}}_\mathcal{S}(\theta_k)\right] > \epsilon\right) \leq \sum_{k=1}^N \left(\frac{en}{v_k}\right)^{v_k} e^{-2n\epsilon^2} \tag{14}$$

$$\leq \sum_{k=1}^N \left(\frac{en}{v_k}\right)^{v_k} e^{-2n\epsilon_k^2} \tag{15}$$

$$= \sum_{k=1}^N \frac{\delta}{N} = \delta, \tag{16}$$

since $\epsilon > \sqrt{\frac{(v_k[\ln(n/v_k)+1]+\ln(N/\delta))}{2n}}$ for all $k$. Hence, the inequality holds with probability at least $1 - \delta$.

Based on this fact, let us consider the set of events such that $\max_k \left[ \mathcal{E}_S(\theta_k) - \hat{\mathcal{E}}_S(\theta_k) \right] \leq \epsilon$. Then, for any $\theta$, there exists $k'$ such that $\theta \in \Theta_{k'}$. Then, we get

$$\mathcal{E}_S(\theta) - \hat{\mathcal{E}}_S^\gamma(\theta) \leq \mathcal{E}_S(\theta) - \hat{\mathcal{E}}_S(\theta_{k'}) \tag{17}$$

$$\leq \mathcal{E}_S(\theta) - \mathcal{E}_S(\theta_{k'}) + \epsilon \tag{18}$$

$$\leq \mathcal{E}_S(\theta_{k'}) - \mathcal{E}_S(\theta_{k'}) + \epsilon = \epsilon, \tag{19}$$

where the second inequality holds since $\mathcal{E}_S(\theta_{k'}) - \hat{\mathcal{E}}_S(\theta_{k'}) \leq \max_k \left[ \mathcal{E}_S(\theta_k) - \hat{\mathcal{E}}_S(\theta_k) \right] \leq \epsilon$ and the final inequality holds since $\theta_{k'}$ is the local maximum in $\Theta_{k'}$. In this regards, we know that $\max_k \left[ \mathcal{E}_S(\theta_k) - \hat{\mathcal{E}}_S(\theta_k) \right] \leq \epsilon$ implies $\mathcal{E}_S(\theta) - \hat{\mathcal{E}}_S^\gamma(\theta) \leq \epsilon$. Consequently, $\mathcal{E}_S(\theta) - \hat{\mathcal{E}}_S^\gamma(\theta) \leq \epsilon$ holds with probability at least $1 - \delta$. $\qquad\square$

## C.2 Proof of Theorem 1

*Proof.* The proof consists of two parts. First, we show that the following inequality holds with high probability.

$$\mathcal{E}_\mathcal{T}(\theta) \leq \hat{\mathcal{E}}_S^\gamma(\theta) + \frac{1}{2}\mathbf{Div}\left(\mathcal{S}, \mathcal{T}\right) + \max_k \sqrt{\frac{(v_k \left[\ln\left(n/v_k\right) + 1\right] + \ln\left(N/\delta\right))}{2n}}.$$

Then, secondly, we apply the inequality for multiple source domains.

The first part can be proven by simply combining Lemma 1 and Lemma 2. Then, we get,

$$\mathcal{E}_\mathcal{T}(\theta) \leq \mathcal{E}_\mathcal{S}(\theta) + \frac{1}{2}\mathbf{Div}\left(\mathcal{S}, \mathcal{T}\right) \tag{20}$$

$$\leq \hat{\mathcal{E}}_S^\gamma(\theta) + \frac{1}{2}\mathbf{Div}\left(\mathcal{S}, \mathcal{T}\right) + \max_k \sqrt{\frac{(v_k \left[\ln\left(n/v_k\right) + 1\right] + \ln\left(N/\delta\right))}{2n}} \tag{21}$$

where $\mathbf{Div}\left(\mathcal{S}, \mathcal{T}\right)$ is a divergence between $\mathcal{S}$ and $\mathcal{T}$.

For the second part, we set $\mathcal{D} := \sum_{i=1}^I \mathcal{D}_i / I$ which is a mixture of source distributions. Then, by applying $\mathcal{D}$ to the first part, we obtain the following inequality,

$$\mathcal{E}_\mathcal{T}(\theta) \leq \hat{\mathcal{E}}_\mathcal{D}^\gamma(\theta) + \frac{1}{2}\mathbf{Div}\left(\mathcal{D}, \mathcal{T}\right) + \max_k \sqrt{\frac{(v_k \left[\ln\left(In/v_k\right) + 1\right] + \ln\left(N/\delta\right))}{2In}} \tag{22}$$

$$\leq \hat{\mathcal{E}}_\mathcal{D}^\gamma(\theta) + \frac{1}{2I}\sum_{i=1}^I \mathbf{Div}\left(\mathcal{D}_i, \mathcal{T}\right) + \max_k \sqrt{\frac{(v_k \left[\ln\left(In/v_k\right) + 1\right] + \ln\left(N/\delta\right))}{2In}} \tag{23}$$

where the total number of training data set is $In$ and, for the second inequality, we use the fact that $\frac{1}{2}\mathbf{Div}\left(\mathcal{D}, \mathcal{T}\right) \leq \frac{1}{2I}\sum_{i=1}^I \mathbf{Div}\left(\mathcal{D}_i, \mathcal{T}\right)$, which has been proven in [14]. $\qquad\square$

## C.3 Proof of Theorem 2

*Proof.* First, let $\bar{\theta} \in \arg\max_{\theta \in \Theta} \mathcal{E}_\mathcal{T}(\theta)$. Then, from generalization error bound of $\mathcal{E}_\mathcal{D}(\bar{\theta})$, the following inequality holds with probability at most $\frac{\delta}{2}$,

$$\hat{\mathcal{E}}_\mathcal{D}(\bar{\theta}) - \mathcal{E}_\mathcal{D}(\bar{\theta}) > \sqrt{\frac{v \ln\left(In/v\right) + \ln\left(2/\delta\right)}{In}}, \tag{24}$$

where $v$ is a VC dimension of $\Theta$. Furthermore, from Theorem 1, we have the following inequality with probability at most $\frac{\delta}{2}$,

$$\mathcal{E}_\mathcal{T}(\hat{\theta}^\gamma) > \mathcal{E}_\mathcal{D}^\gamma(\hat{\theta}^\gamma) + \frac{1}{2}\mathbf{Div}(\mathcal{D}, \mathcal{T}) + \max_{k \in [1,N]} \sqrt{\frac{v_k \ln\left(In/v_k\right) + \ln(2N/\delta)}{In}}. \tag{25}$$

Finally, let us consider the set of event such that $\hat{\mathcal{E}}_\mathcal{D}(\bar{\theta}) - \mathcal{E}_\mathcal{D}(\bar{\theta}) \leq \sqrt{\frac{v \ln(In/v) + \ln(2/\delta)}{In}}$ and $\mathcal{E}_\mathcal{T}(\hat{\theta}^\gamma) \leq \mathcal{E}_\mathcal{D}^\gamma(\hat{\theta}^\gamma) + \frac{1}{2}\mathbf{Div}(\mathcal{D}, \mathcal{T}) + \max_{k \in [1,N]} \sqrt{\frac{v_k \ln(In/v_k) + \ln(2N/\delta)}{In}}$ whose probability is at least greater

than $1 - \delta$. Then, under this set of event, we have,

$$\min_{\theta'} \hat{\mathcal{E}}_{\mathcal{D}}(\theta') \leq \hat{\mathcal{E}}_{\mathcal{D}}(\bar{\theta}) \leq \mathcal{E}_{\mathcal{D}}(\bar{\theta}) + \sqrt{\frac{v \ln (In/v) + \ln (2/\delta)}{In}} \tag{26}$$

$$\leq \mathcal{E}_{\mathcal{T}}(\bar{\theta}) + \frac{1}{2}\mathbf{Div}(\mathcal{D}, \mathcal{T}) + \sqrt{\frac{v \ln (In/v) + \ln (2/\delta)}{In}} \tag{27}$$

$$\leq \min_{\theta'} \mathcal{E}_{\mathcal{T}}(\theta') + \frac{1}{2}\mathbf{Div}(\mathcal{D}, \mathcal{T}) + \sqrt{\frac{v \ln (In/v) + \ln (2/\delta)}{In}} \tag{28}$$

Consequently, we have,

$$\mathcal{E}_{\mathcal{T}}(\hat{\theta}^{\gamma}) - \min_{\theta'} \mathcal{E}_{\mathcal{T}}(\theta')$$

$$\leq \mathcal{E}_{\mathcal{D}}^{\gamma}(\hat{\theta}^{\gamma}) - \min_{\theta'} \hat{\mathcal{E}}_{\mathcal{D}}(\theta') + \mathbf{Div}(\mathcal{D}, \mathcal{T}) + \max_{k \in [1,N]} \sqrt{\frac{v_k \ln (In/v_k) + \ln(2N/\delta)}{In}}$$

$$+ \sqrt{\frac{v \ln (In/v) + \ln (2/\delta)}{In}} \tag{29}$$

$$\leq \mathcal{E}_{\mathcal{D}}^{\gamma}(\hat{\theta}^{\gamma}) - \min_{\theta'} \hat{\mathcal{E}}_{\mathcal{D}}(\theta') + \frac{1}{I} \sum_{i=1}^{I} \mathbf{Div}(\mathcal{D}_i, \mathcal{T}) + \max_{k \in [1,N]} \sqrt{\frac{v_k \ln (In/v_k) + \ln(2N/\delta)}{In}}$$

$$+ \sqrt{\frac{v \ln (In/v) + \ln (2/\delta)}{In}} \tag{30}$$

$\square$

# D   Additional Experiments

## D.1   Comparison of flatness-aware solvers

Table 2: **Flatness-aware solvers comparison.** SWAs collect 10 weights from the last 20% of training.

| Algorithm | PACS | VLCS | OfficeHome | TerraInc | DomainNet | Avg. |
|---|---|---|---|---|---|---|
| ERM (baseline) | 85.5 ±0.2 | 77.5 ±0.4 | 66.5 ±0.3 | 46.1 ±1.8 | 40.9 ±0.1 | 63.3 |
| SAM | 85.8 ±0.2 | **79.4** ±0.1 | 69.6 ±0.1 | 43.3 ±0.7 | 44.3 ±0.0 | 64.5 |
| SWA$_{\text{w/ cyclic}}$ | 87.1 ±0.1 | 76.5 ±0.2 | 68.5 ±0.2 | 49.6 ±1.0 | 45.6 ±0.0 | 65.5 |
| SWA$_{\text{w/ const}}$ | 86.9 ±0.2 | 76.6 ±0.1 | 69.3 ±0.3 | 49.2 ±1.2 | 45.9 ±0.0 | 65.6 |
| SWAD | **88.1** ±0.1 | 79.1 ±0.1 | **70.6** ±0.2 | **50.0** ±0.3 | **46.5** ±0.1 | **66.9** |

Interestingly, the average performance ranking of flatness-aware solvers is the same as the results of the local flatness test (See Figure 3 in the main text). In both experiments, SWAD performs best, followed by SWAs, SAM, and ERM. It is another evidence of our claim that domain generalization is achievable by seeking flat minima.

On the other hand, comparing SWAs and SWAD demonstrates the effectiveness of the proposed dense and overfit-aware sampling strategy. SWAD improves average performance up to 1.4pp, and surpasses both SWAs on every benchmark.

# E   Full Results

In this section, we show detailed results of Table 2 in the main text. † and ‡ indicate results from DomainBed's and our HP search protocols, respectively. Standard errors are reported from three trials, if available.

## E.1 PACS

Table 3: **Out-of-domain accuracies (%) on** PACS.

| Algorithm | A | C | P | S | Avg |
|---|---|---|---|---|---|
| CDANN[†] | 84.6 ±1.8 | 75.5 ±0.9 | 96.8 ±0.3 | 73.5 ±0.6 | 82.6 |
| MASF | 82.9 | 80.5 | 95.0 | 72.3 | 82.7 |
| DMG | 82.6 | 78.1 | 94.5 | 78.3 | 83.4 |
| IRM[†] | 84.8 ±1.3 | 76.4 ±1.1 | 96.7 ±0.6 | 76.1 ±1.0 | 83.5 |
| MetaReg | 87.2 | 79.2 | 97.6 | 70.3 | 83.6 |
| DANN[†] | 86.4 ±0.8 | 77.4 ±0.8 | 97.3 ±0.4 | 73.5 ±2.3 | 83.7 |
| ERM[‡] | 85.7 ±0.6 | 77.1 ±0.8 | 97.4 ±0.4 | 76.6 ±0.7 | 84.2 |
| GroupDRO[†] | 83.5 ±0.9 | 79.1 ±0.6 | 96.7 ±0.3 | 78.3 ±2.0 | 84.4 |
| MTL[†] | 87.5 ±0.8 | 77.1 ±0.5 | 96.4 ±0.8 | 77.3 ±1.8 | 84.6 |
| I-Mixup | 86.1 ±0.5 | 78.9 ±0.8 | 97.6 ±0.1 | 75.8 ±1.8 | 84.6 |
| MMD[†] | 86.1 ±1.4 | 79.4 ±0.9 | 96.6 ±0.2 | 76.5 ±0.5 | 84.7 |
| VREx[†] | 86.0 ±1.6 | 79.1 ±0.6 | 96.9 ±0.5 | 77.7 ±1.7 | 84.9 |
| MLDG[†] | 85.5 ±1.4 | 80.1 ±1.7 | 97.4 ±0.3 | 76.6 ±1.1 | 84.9 |
| ARM[†] | 86.8 ±0.6 | 76.8 ±0.5 | 97.4 ±0.3 | 79.3 ±1.2 | 85.1 |
| RSC[†] | 85.4 ±0.8 | 79.7 ±1.8 | 97.6 ±0.3 | 78.2 ±1.2 | 85.2 |
| Mixstyle[‡] | 86.8 ±0.5 | 79.0 ±1.4 | 96.6 ±0.1 | 78.5 ±2.3 | 85.2 |
| ER | 87.5 | 79.3 | **98.3** | 76.3 | 85.3 |
| pAdaIN | 85.8 | 81.1 | 97.2 | 77.4 | 85.4 |
| ERM[†] | 84.7 ±0.4 | 80.8 ±0.6 | 97.2 ±0.3 | 79.3 ±1.0 | 85.5 |
| EISNet | 86.6 | 81.5 | 97.1 | 78.1 | 85.8 |
| CORAL[†] | 88.3 ±0.2 | 80.0 ±0.5 | 97.5 ±0.3 | 78.8 ±1.3 | 86.2 |
| SagNet[†] | 87.4 ±1.0 | 80.7 ±0.6 | 97.1 ±0.1 | 80.0 ±0.4 | 86.3 |
| DSON | 87.0 | 80.6 | 96.0 | **82.9** | 86.6 |
| Ours | **89.3** ±0.2 | **83.4** ±0.6 | 97.3 ±0.3 | 82.5 ±0.5 | **88.1** |

## E.2 VLCS

Table 4: **Out-of-domain accuracies (%) on** VLCS.

| Algorithm | C | L | S | V | Avg |
|---|---|---|---|---|---|
| GroupDRO[†] | 97.3 ±0.3 | 63.4 ±0.9 | 69.5 ±0.8 | 76.7 ±0.7 | 76.7 |
| RSC[†] | 97.9 ±0.1 | 62.5 ±0.7 | 72.3 ±1.2 | 75.6 ±0.8 | 77.1 |
| MLDG[†] | 97.4 ±0.2 | 65.2 ±0.7 | 71.0 ±1.4 | 75.3 ±1.0 | 77.2 |
| MTL[†] | 97.8 ±0.4 | 64.3 ±0.3 | 71.5 ±0.7 | 75.3 ±1.7 | 77.2 |
| ERM[‡] | 98.0 ±0.3 | 64.7 ±1.2 | 71.4 ±1.2 | 75.2 ±1.6 | 77.3 |
| I-Mixup | 98.3 ±0.6 | 64.8 ±1.0 | 72.1 ±0.5 | 74.3 ±0.8 | 77.4 |
| ERM[†] | 97.7 ±0.4 | 64.3 ±0.9 | 73.4 ±0.5 | 74.6 ±1.3 | 77.5 |
| MMD[†] | 97.7 ±0.1 | 64.0 ±1.1 | 72.8 ±0.2 | 75.3 ±3.3 | 77.5 |
| CDANN[†] | 97.1 ±0.3 | 65.1 ±1.2 | 70.7 ±0.8 | 77.1 ±1.5 | 77.5 |
| ARM[†] | 98.7 ±0.2 | 63.6 ±0.7 | 71.3 ±1.2 | 76.7 ±0.6 | 77.6 |
| SagNet[†] | 97.9 ±0.4 | 64.5 ±0.5 | 71.4 ±1.3 | 77.5 ±0.5 | 77.8 |
| Mixstyle[‡] | 98.6 ±0.3 | 64.5 ±1.1 | 72.6 ±0.5 | 75.7 ±1.7 | 77.9 |
| VREx[†] | 98.4 ±0.3 | 64.4 ±1.4 | 74.1 ±0.4 | 76.2 ±1.3 | 78.3 |
| IRM[†] | 98.6 ±0.1 | 64.9 ±0.9 | 73.4 ±0.6 | 77.3 ±0.9 | 78.6 |
| DANN[†] | **99.0** ±0.3 | 65.1 ±1.4 | 73.1 ±0.3 | 77.2 ±0.6 | 78.6 |
| CORAL[†] | 98.3 ±0.1 | **66.1** ±1.2 | 73.4 ±0.3 | 77.5 ±1.2 | 78.8 |
| Ours | 98.8 ±0.1 | 63.3 ±0.3 | **75.3** ±0.5 | **79.2** ±0.6 | **79.1** |

## E.3 OfficeHome

Table 5: **Out-of-domain accuracies (%) on** `OfficeHome`.

| Algorithm | A | C | P | R | Avg |
|---|---|---|---|---|---|
| Mixstyle[‡] | 51.1 ±0.3 | 53.2 ±0.4 | 68.2 ±0.7 | 69.2 ±0.6 | 60.4 |
| IRM[†] | 58.9 ±2.3 | 52.2 ±1.6 | 72.1 ±2.9 | 74.0 ±2.5 | 64.3 |
| ARM[†] | 58.9 ±0.8 | 51.0 ±0.5 | 74.1 ±0.1 | 75.2 ±0.3 | 64.8 |
| RSC[†] | 60.7 ±1.4 | 51.4 ±0.3 | 74.8 ±1.1 | 75.1 ±1.3 | 65.5 |
| CDANN[†] | 61.5 ±1.4 | 50.4 ±2.4 | 74.4 ±0.9 | 76.6 ±0.8 | 65.7 |
| DANN[†] | 59.9 ±1.3 | 53.0 ±0.3 | 73.6 ±0.7 | 76.9 ±0.5 | 65.9 |
| GroupDRO[†] | 60.4 ±0.7 | 52.7 ±1.0 | 75.0 ±0.7 | 76.0 ±0.7 | 66.0 |
| MMD[†] | 60.4 ±0.2 | 53.3 ±0.3 | 74.3 ±0.1 | 77.4 ±0.6 | 66.4 |
| MTL[†] | 61.5 ±0.7 | 52.4 ±0.6 | 74.9 ±0.4 | 76.8 ±0.4 | 66.4 |
| VREx[†] | 60.7 ±0.9 | 53.0 ±0.9 | 75.3 ±0.1 | 76.6 ±0.5 | 66.4 |
| ERM[†] | 61.3 ±0.7 | 52.4 ±0.3 | 75.8 ±0.1 | 76.6 ±0.3 | 66.5 |
| MLDG[†] | 61.5 ±0.9 | 53.2 ±0.6 | 75.0 ±1.2 | 77.5 ±0.4 | 66.8 |
| ERM[‡] | 63.1 ±0.3 | 51.9 ±0.4 | 77.2 ±0.5 | 78.1 ±0.2 | 67.6 |
| I-Mixup | 62.4 ±0.8 | 54.8 ±0.6 | 76.9 ±0.3 | 78.3 ±0.2 | 68.1 |
| SagNet[†] | 63.4 ±0.2 | 54.8 ±0.4 | 75.8 ±0.4 | 78.3 ±0.3 | 68.1 |
| CORAL[†] | 65.3 ±0.4 | 54.4 ±0.5 | 76.5 ±0.1 | 78.4 ±0.5 | 68.7 |
| Ours | **66.1** ±0.4 | **57.7** ±0.4 | **78.4** ±0.1 | **80.2** ±0.2 | **70.6** |

## E.4 TerraIncognita

Table 6: **Out-of-domain accuracies (%) on** `TerraIncognita`.

| Algorithm | L100 | L38 | L43 | L46 | Avg |
|---|---|---|---|---|---|
| MMD[†] | 41.9 ±3.0 | 34.8 ±1.0 | 57.0 ±1.9 | 35.2 ±1.8 | 42.2 |
| GroupDRO[†] | 41.2 ±0.7 | 38.6 ±2.1 | 56.7 ±0.9 | 36.4 ±2.1 | 43.2 |
| Mixstyle[‡] | 54.3 ±1.1 | 34.1 ±1.1 | 55.9 ±1.1 | 31.7 ±2.1 | 44.0 |
| ARM[†] | 49.3 ±0.7 | 38.3 ±2.4 | 55.8 ±0.8 | 38.7 ±1.3 | 45.5 |
| MTL[†] | 49.3 ±1.2 | 39.6 ±6.3 | 55.6 ±1.1 | 37.8 ±0.8 | 45.6 |
| CDANN[†] | 47.0 ±1.9 | 41.3 ±4.8 | 54.9 ±1.7 | 39.8 ±2.3 | 45.8 |
| ERM[†] | 49.8 ±4.4 | 42.1 ±1.4 | 56.9 ±1.8 | 35.7 ±3.9 | 46.1 |
| VREx[†] | 48.2 ±4.3 | 41.7 ±1.3 | 56.8 ±0.8 | 38.7 ±3.1 | 46.4 |
| RSC[†] | 50.2 ±2.2 | 39.2 ±1.4 | 56.3 ±1.4 | **40.8** ±0.6 | 46.6 |
| DANN[†] | 51.1 ±3.5 | 40.6 ±0.6 | 57.4 ±0.5 | 37.7 ±1.8 | 46.7 |
| IRM[†] | 54.6 ±1.3 | 39.8 ±1.9 | 56.2 ±1.8 | 39.6 ±0.8 | 47.6 |
| CORAL[†] | 51.6 ±2.4 | 42.2 ±1.0 | 57.0 ±1.0 | 39.8 ±2.9 | 47.7 |
| MLDG[†] | 54.2 ±3.0 | 44.3 ±1.1 | 55.6 ±0.3 | 36.9 ±2.2 | 47.8 |
| I-Mixup | **59.6** ±2.0 | 42.2 ±1.4 | 55.9 ±0.8 | 33.9 ±1.4 | 47.9 |
| SagNet[†] | 53.0 ±2.9 | 43.0 ±2.5 | 57.9 ±0.6 | 40.4 ±1.3 | 48.6 |
| ERM[‡] | 54.3 ±0.4 | 42.5 ±0.7 | 55.6 ±0.3 | 38.8 ±2.5 | 47.8 |
| Ours | 55.4 ±0.0 | **44.9** ±1.1 | **59.7** ±0.4 | 39.9 ±0.2 | **50.0** |

### E.5 DomainNet

Table 7: **Out-of-domain accuracies (%) on** DomainNet.

| Algorithm | clip | info | paint | quick | real | sketch | Avg |
|---|---|---|---|---|---|---|---|
| MMD[†] | 32.1 ±13.3 | 11.0 ±4.6 | 26.8 ±11.3 | 8.7 ±2.1 | 32.7 ±13.8 | 28.9 ±11.9 | 23.4 |
| GroupDRO[†] | 47.2 ±0.5 | 17.5 ±0.4 | 33.8 ±0.5 | 9.3 ±0.3 | 51.6 ±0.4 | 40.1 ±0.6 | 33.3 |
| VREx[†] | 47.3 ±3.5 | 16.0 ±1.5 | 35.8 ±4.6 | 10.9 ±0.3 | 49.6 ±4.9 | 42.0 ±3.0 | 33.6 |
| IRM[†] | 48.5 ±2.8 | 15.0 ±1.5 | 38.3 ±4.3 | 10.9 ±0.5 | 48.2 ±5.2 | 42.3 ±3.1 | 33.9 |
| Mixstyle[‡] | 51.9 ±0.4 | 13.3 ±0.2 | 37.0 ±0.5 | 12.3 ±0.1 | 46.1 ±0.3 | 43.4 ±0.4 | 34.0 |
| ARM[†] | 49.7 ±0.3 | 16.3 ±0.5 | 40.9 ±1.1 | 9.4 ±0.1 | 53.4 ±0.4 | 43.5 ±0.4 | 35.5 |
| CDANN[†] | 54.6 ±0.4 | 17.3 ±0.1 | 43.7 ±0.9 | 12.1 ±0.7 | 56.2 ±0.4 | 45.9 ±0.5 | 38.3 |
| DANN[†] | 53.1 ±0.2 | 18.3 ±0.1 | 44.2 ±0.7 | 11.8 ±0.1 | 55.5 ±0.4 | 46.8 ±0.6 | 38.3 |
| RSC[†] | 55.0 ±1.2 | 18.3 ±0.5 | 44.4 ±0.6 | 12.2 ±0.2 | 55.7 ±0.7 | 47.8 ±0.9 | 38.9 |
| I-Mixup | 55.7 ±0.3 | 18.5 ±0.5 | 44.3 ±0.5 | 12.5 ±0.4 | 55.8 ±0.3 | 48.2 ±0.5 | 39.2 |
| SagNet[†] | 57.7 ±0.3 | 19.0 ±0.2 | 45.3 ±0.3 | 12.7 ±0.5 | 58.1 ±0.5 | 48.8 ±0.2 | 40.3 |
| MTL[†] | 57.9 ±0.5 | 18.5 ±0.4 | 46.0 ±0.1 | 12.5 ±0.1 | 59.5 ±0.3 | 49.2 ±0.1 | 40.6 |
| ERM[†] | 58.1 ±0.3 | 18.8 ±0.3 | 46.7 ±0.3 | 12.2 ±0.4 | 59.6 ±0.1 | 49.8 ±0.4 | 40.9 |
| MLDG[†] | 59.1 ±0.2 | 19.1 ±0.3 | 45.8 ±0.7 | 13.4 ±0.3 | 59.6 ±0.2 | 50.2 ±0.4 | 41.2 |
| CORAL[†] | 59.2 ±0.1 | 19.7 ±0.2 | 46.6 ±0.3 | 13.4 ±0.4 | 59.8 ±0.2 | 50.1 ±0.6 | 41.5 |
| MetaReg | 59.8 | **25.6** | 50.2 | 11.5 | 64.6 | 50.1 | 43.6 |
| DMG | 65.2 | 22.2 | 50.0 | 15.7 | 59.6 | 49.0 | 43.6 |
| ERM[‡] | 63.0 ±0.2 | 21.2 ±0.2 | 50.1 ±0.4 | 13.9 ±0.5 | 63.7 ±0.2 | 52.0 ±0.5 | 44.0 |
| Ours | **66.0** ±0.1 | 22.4 ±0.3 | **53.5** ±0.1 | **16.1** ±0.2 | **65.8** ±0.4 | **55.5** ±0.3 | **46.5** |

## F  Assets

In this section, we discuss about licenses, copyrights, and ethical issues of our assets, such as code and datasets.

### F.1  Code

Our work is built upon DomainBed [1][2], which is released under the MIT license.

### F.2  Datasets

While we use public datasets only, we track how the datasets were built to discuss licenses, copyrights, and potential ethical issues. For DomainNet [15] and OfficeHome [16], we use the datasets for non-profit academic research only following their fair use notice. TerraIncognita [17] is a subset of Caltech Camera Traps (CCT) dataset, distributed under the Community Data License Agreement (CDLA) license. PACS [18] and VLCS [19] datasets have images collected from the web and we could not find any statements about licenses, copyrights, or whether consent was obtained. Considering that both datasets contain person class and images of people, there may be potential ethical issues.

## G  Reproducibility

To provide details of our algorithm and guarantee reproducibility, we provide the source code[3] publicly. The code also specifies detailed environments, dependencies, how to download datasets, and instructions to reproduce the main results (Table 1 and 2 in the main text).

---

[2]https://github.com/facebookresearch/DomainBed
[3]https://github.com/khanrc/swad

### G.1 Infrastructures

Every experiment is conducted on a single NVIDIA Tesla P40 or V100, Python 3.8.6, PyTorch 1.7.0, Torchvision 0.8.1, and CUDA 9.2.

### G.2 Runtime Analysis

The total runtime varies depending on datasets and the moment detected to overfit. It takes about 4 hours for `PACS` and `VLCS`, 8 hours for `OfficeHome`, 8.5 hours for `TerraIncognita`, and 56 hours for `DomainNet` on average, when using a single NVIDIA Tesla P40 GPU. Each experiment includes the leave-one-out cross-validations for all domains in each dataset.

### G.3 Complexity Analysis

The only additional time overhead incurs from stochastic weights selection, which requires further evaluations. To analyze the overhead, let the forward time $t_f$, backward time $t_b$, training and validation split ratio $r = |X^{train}|/|X^{valid}|$, total in-domain samples $n$, and evaluation frequency $v$ that indicates how many evaluations are conducted for each epoch. For conciseness, we assume $t = t_f = t_b$ and do not consider early stopping.

For one epoch, training time is $2tnr/(r+1)$, and evaluation time is $vtn/(r+1)$. The total runtime for one epoch is $tn(2r+v)/(r+1)$. Final overhead ratio is $(2r+v)/(2r+v_b)$ where $v_b$ is the evaluation frequency of a baseline. In our main experiments, we use $r = 4$. Compared to the default parameters of DomainBed [1], we use $v = 2v_b$ for `DomainNet`, $v = 6v_b$ for `VLCS`, and $v = 3v_b$ for the others. Then, the total runtime of our algorithm takes from 1.07 (`PACS`) to 1.27 (`DomainNet`) times more than the ERM baseline. In practice, it can be improved by conducting approximated evaluations using sub-sampled validation set.

In terms of memory complexity, our method does not require additional GPU memory. Instead, we leverage CPU memory to minimize training time overhead, which takes up to $\max(N, M)$ times more than the baseline.