# OpenReview forum: "SWAD: Domain Generalization by Seeking Flat Minima"
_NeurIPS.cc/2021/Conference — NeurIPS 2021 Poster_

### Official Review · Reviewer_1Nc7 · 2021-06-29

**Rating:** 6
**Confidence:** 4

**Summary:**

This paper extends the idea of Stochastic Weighting Averaging (SWA) to the context of Domain Generalization (DG). Theoretically, the authors argues that finding a flat minima improves domain generalization. Algorithmically, the authors modify SWA by increasing the number of samples being averaged (e.g. per iteration) and by selecting the interval from where the samples are taken. The proposed algorithm  is evaluated in  the DomainBed benchmark and shows compelling results.

**Limitations And Societal Impact:**

Yes.

**Main Review:**

Pros:

- The paper tackles an important and relevant problem to the ML community.
- The proposed algorithm is relatively simple to implement in practice and perform well experimentally.

Cons:

- My major concern is wrt to the novelty of the work and the connection/motivation between the proposed algorithm SWAD, domain generalization and the derived theoretical insights.

- __Theory__.  While I appreciate the  provided theoretical insights, they directly follow from standard results in domain adaptation [1,2,3]  and Norton and Royset [29] . Most importantly, I believe some details  have been overseen in the technical proof.  Specifically, I believe the proof misses a term  (line 86 eq. 20 in suppl material).  Lemma 1 is between any two h, but the risk is between h and the labeling function. If I understand correct, this is overseen in line 86.  The author should rewrite lemma 1 before directly using it in the proof. Please refer to [1] and [2] for details. The second part of the proof and Lemma 2 follow from Norton and Royset [5].  This should be clearly mentioned and if they are different the authors should highlight main differences.  I would appreciate if the authors can clarify on this.

    Moreover, in practice,  the provided upper bound can be easily dominated by the divergence between  source and target.  Theorem 1, does not capture either the dissimilarity of the labeling functions (as a result of the missing term), which can also simply dominate the performance on target and being small is necessary for adaptation [2,3,4].  Even if these were all negligible and the result correct, the provided bound only shows that a flat minimal can  improve generalization but this is not necessarily novel.   It is hard to see a clear connection between these results and the proposed algorithm.  In other words, the theoretical results do not necessarily motivate the use of SWAD vs SWA. I would appreciate if the authors can provide further explanation on this aspect.

- __Algorithm__. The proposed algorithm is based on a heuristic on top of SWA  that does not necessarily exploit the current scenario (DG). I believe this heuristic improves SWA in general and it might be algorithmically incremental vs vanilla SWA. Moreover, it introduces 3 new hyperparameters (well tuned in the experiments).  A better motivation than the one provided in 134-147 can improve the presentation/motivation of the algorithm.

- __Experiments and Analysis__.  This is one of the strongest sections of the paper.  Just a few questions:
    - Why SWA is not shown in Table 2?
    - Table 4 mentioned HP search was not applied but ERM+SWAD includes HP search (This is the same result from Table 2).

__Suppl__

- Lemma 1 is a well know result also from [1,2,3] and should be cited appropriately.
- Lemma 2 follows [5] and show be cited appropriately.






**Originality**: The use of SWA in the context of DG is new.  However, algorithmically the proposed method is an extension (based on a heuristic) of a well known technique.  Theoretically, the presented results also follow from existing work and it is not clear how they motivate the need of SWA vs SWAD (see theory and algorithm above for more details).

**Quality**:  I believe the proposed algorithm should be better motivated as it is not clear the connection/motivation between  SWAD, domain generalization and the derived theoretical insights.  Moreover,  I believe several  small details where overseen in the technical proofs and those should be corrected to make them technically sound. (theory above for more details)

**Clarity**:  The submission is clearly written and well organized. The main issue in this aspect is with respect to the motivation and connection among the two different parts (theoretical results and algorithms).  (see algorithm above for more details)

**Significance**:  The empirical section is one of the strongest part of the paper and the results advance SoTA.  The method is relatively simple to implement in practice. However, it introduces a few extra hyperparams.

As expressed above, my major concern is wrt to the novelty of the work and the connection between the proposed algorithm SWAD, domain generalization and the derived theoretical insights. I believe the paper could significantly benefit  from a major revision.


__References__

[1] Ben-David et al.  A theory of learning from different domains Springer 2010

[2] Impossibility Theorems for Domain Adaptation PMLR,2010.

[3] Acuna et al. f-Domain-Adversarial Learning: Theory and Algorithms. ICML 2021

[4] Zhao et al On Learning Invariant Representations for Domain Adaptation, ICML 2019

[5] Norton et al Diametrical Risk Minimization: Theory and Computations, 2019 (cited)

**Time Spent Reviewing:**

11

---

> ### Author Response · Authors · 2021-08-10
> **Author response for Reviewer 1Nc7**
>
> We deeply appreciate the valuable comments by Reviewer 1Nc7. We will address all concerns raised by the reviewer.
>
> ### Novelty of the work & Connections between SWAD, DG, and the derived theoretical insights
> The main contribution of this study is (1) first introducing theoretical results on flatness into the domain generalization setting -- Section 2, and (2) proposing an empirically well-performed approximated flatness-aware solver, named SWAD -- Section 3.
>
> We first introduce the relationship between flatness (robust risk) and domain generalization (DG), which is not yet explored in the community. Our theoretical results introduce the concept of flatness (i.e., the robust risk minimization) into the results of [A] (i.e., into the DG tasks). We also make a theoretical connection between the empirical robust risk solution (what we are arguing to solve), the empirical risk solution (what previous works have focused on), and the true domain generalization gap (what we are truly interested in). Note that theorems 1 and 2 are not novel by themselves (particularly, compared to [A]). However, our theoretical novelty is introducing the theoretical foundation of bringing the concept of flatness into DG.
>
> From the theoretical observation, we applied several flatness-aware solvers, such as SWA [25] or SAM [26]. We also tested SAM [26], but we observed that SWA practically works better than SAM, in terms of both computational cost (SAM requires twice the cost) and empirical performance (64.5% for SAM / 65.5% for SWA).
>
> At the same time, we also observed that directly applying SWA to DG tasks raises two problems; (1) not enough number of gathered weights (2) overfitting. We fix the vanilla SWA to mitigate the issues, and show the empirical benefit by our proposed method (We will discuss the hyperparameter concerns in the later section). As a result, our SWAD shows +1.4pp improvements compared to the vanilla SWA and +3.6pp improvements compared to the vanilla ERM. SWA(D) is not theoretically guaranteed to reduce robust risk, but the empirical results (Fig 3, and [25]) show it finds flat minima well. We do not think our modification is significantly novel (​​we did not focus on developing a perfect flatness-aware solver as discussed in Section 5), but we believe that our technical contribution is making the vanilla SWA practical and stable for DG tasks.
>
> Furthermore, as shown in Table 3, the well-known and widely-applicable in-distribution generalization methods, such as Mixup, CutMix, VAT, cannot improve the DG performances, while they consistently show better in-distribution generalization performances. In contrast, as shown in Table 3 and 4, flatness-aware methods (SWA, SAM, SWAD) show consistent performance improvements against the ERM counterpart.
>
> As we empirically observed in Table 4, SWAD can be easily combined with other DG methods, e.g., CORAL, and SWAD + CORAL achieves the best performance over all previous comparison methods. We believe our theoretical and empirical contributions open up new domain generalization researches towards state-of-the-arts (upon our method -- the current SOTA) and related theoretical works.
>
> Even if one proposes a better flatness-aware solver than SWAD, our contribution is not harmed, because we believe our main contribution is bringing the concept of flatness into DG tasks as discussed in Section 5.
>
> We hope that the reviewer feels that this response is a better motivation than L134-147. We will revise the paper accordingly.
>
> ### [Theory] Missed component in the proof (line 86 eq. 20 in appendix) & Theorem 1, does not capture either the dissimilarity of the labeling functions (..) necessary for adaptation [B, C, D]
> We really appreciate the reviewer carefully checking our proofs. We apologize for any inconvenience to the reviewer due to the missing assumptions for the labeling function in our theories. We will clearly state the assumptions of the labeling function.
>
> In our theory, we assume there exists a global labeling function, i.e., target and source labeling functions are the same. We also assume that the global labeling function is included in the hypothesis set. These assumptions make our bound different from [A] and [B]. The generalization bound in [A] and [B] contains the term, $\lambda^*$, the optimal joint risk that can be achieved by the hypothesis class. However, because we assume that the global labeling function is in the hypothesis set, the optimal joint risk becomes zero. For this reason, we omitted the term, $\lambda^*$, when we apply Lemma 1 to Theorem 1 (line 86 in Appendix).
>
> Additionally, we omitted the dissimilarity term between labeling functions since the target and source labeling functions are the same. We presume that the reviewer considered $\lambda^*$ and dissimilarity term are missing in our theorem (compared to [A, B]), but we assume different assumptions with [A, B], resulting in the different upper bound with fewer terms. We will clearly state the condition in the paper.
>
> If we consider the same scenario of [A] and [B], the term related to the optimal joint risk or dissimilarity will appear. We believe that it can be easily proven by using the same techniques in [A, B, C, D] because the derivation of the dissimilarity term (or the optimal joint risk) is independent and compatible with our main proof of RRM bound. We will add related discussions in the paper.
>
> ### [Theory] The second part of proof and Lemma 2
> As the reviewer has recognized, we followed [29] by using a finite cover of $\theta$. The difference lies in the usage of VC dimension. It is mentioned in the main paper but missed in Appendix. We will revise Appendix.
>
> ### [Theory] Generalization bound can be dominated by the divergence between source and target.
> The domain divergence term is a common assumption in theoretical generalization bounds. A number of previous DG works have tried to minimize the domain divergence term in the feature-level by applying domain agnostic feature learning, e.g., DANN [9] or CORAL [36]. In this case, our theorem is also applicable to the methods reducing the domain divergence [9, 36]. As we empirically observed in Table 4 that the combination of the flatness-aware solver (SWAD) and the domain divergence reduction method (CORAL) achieved the best performance over all comparison methods.
>
> ### SWAD is incremental extension of SWA and it does not fully exploit domain generalization scenario; it introduces 3 new hyperparameters (well tuned in the experiments)
> We do not argue that SWAD is a perfect solver for seeking flat minima in DG scenario. But we are trying to find a practical and working solution in the DG scenario. In the process of finding flat minima through SWA in DG tasks, we observed two problems: (1) not enough number of gathered weights (2) overfitting. We fix the vanilla SWA to mitigate the issues, and show the empirical benefit by our proposed method.
>
> We additionally conduct experiments on SWA under the same protocol to show the effectiveness of our proposed modification (as well as the request by the reviewer). As a result, our SWAD shows +1.4pp improvements compared to the vanilla SWA and +3.6pp improvements compared to the vanilla ERM. Our technical contribution is making the vanilla SWA practical and stable for DG tasks.
>
> | Method | PACS | VLCS | OfficeHome | TerraInc | DomainNet | Avg |
> |---|---|---|---|---|---|---|
> | SWA    | 87.1 | 76.5 | 68.5 | 49.6 | 45.6 | 65.5 |
> | SWAD | 88.1 | 79.1 | 70.6 | 50.0 | 46.5 | 66.9 |
>
> Also, the hyperparameters (HP) of SWAD are not directly tuned on the test set. We thoroughly followed the HP tuning protocol of DomainBed, which introduces a very strict protocol for a fair comparison; only training domains are used for HP tuning and the test domain is only used for the evaluation, and the final performance is the average over evaluations on all different test domains. Therefore, our achievement is not from the additional HP tuning, but solving the problems we observed.
>
> We conducted additional experiments on testing the robustness to the HP selection of our method; First, we choose a single HP setting and use it for every run; each run consists of different dataset, target domain, and data split. It only shows 0.1pp performance drop on average (66.9% -> 66.8%). Second, we tested many HP combinations on PACS. The results are 87.9% ± 0.3% for 8 combinations (each combination is evaluated by three runs). The results support the robustness to HP selection of our method. The detailed results and discussion will be added to the paper.
>
> Furthermore, our method can be easily combined with methods that exploit domain generalization scenario. We observed that SWAD + CORAL achieved the SOTA score (in Table 4).
>
> Overall, (1) we did not directly tune HP to achieve better performances on test set (2) we observe that SWAD is less sensitive to HP selection (3) SWAD consistently outperforms the vanilla SWA under the fair HP selection protocol (4) SWAD is easily combined with methods exploiting domain labels.
>
> ### Experiments
> #### **Why SWA is not in Table 2?**
> Table 2 compares only DG methods. Comparison with SWA is provided in Table 3 and 5. The full table of SWA is in the above. We will add it to the main paper.
>
> #### **HP search for ERM+SWAD?**
> Thanks for pointing out incorrect description. We meant that when we apply SWAD to CORAL and SAM, we use the same HP as “SWAD”. We will clarify the details.
>
> ### References
> [A] Ben-David et al. A theory of learning from different domains. Springer 2010
> [B] Ben-David et al. Impossibility Theorems for Domain Adaptation. PMLR 2010
> [C] Acuna et al. f-Domain-Adversarial Learning: Theory and Algorithms. ICML 2021
> [D] Zhao et al. On Learning Invariant Representations for Domain Adaptation. ICML 2019

---

> > ### Comment · Reviewer_1Nc7 · 2021-08-18
> > **Thank you for the detailed response.**
> >
> > ***Connection between theory SWAD and DG***.
> >
> > As stated  in my initial review, I appreciate the effort in the provided theoretical insights, however, I still belief the theoretical section does not motivate the proposed algorithm. After reading the authors response, it seems the proposed algorithm was indeed motivated based on empirical observations.  That should be the motivation of the work.
> > Furthermore, the theoretical results on its own are not particularly insightful.
> >
> > ***Correctness of the technical results***
> >
> > I thank the authors for clarifying and acknowledging that assumptions made in the technical proofs were missing.
> >
> > After reading the authors response and looking at the paper again,   I believe the assumption the authors claim to be making in the rebuttal  are strong and restrictive  particularly in the DG scenario.
> > Moreover, those assumptions are not stated anywhere and without clearing stating these, the technical results  are not correct.
> >
> > Furthermore, I do not completely agree with the fact that  those assumptions make the proposed  bounds different from [A] and [B].  The assumptions simple make them simpler and look different.
> >
> > Unless there is an intuition behind those assumptions(and a clear need), I would recommend the authors to avoid them.
> >
> > I still believe the paper could significantly benefit from a major revision, where the major focus is put on the empirical section, and the theory is kept very minor.

---

> > > ### Author Response · Authors · 2021-08-23
> > > **Thank you for the quick response**
> > >
> > > **Connection between theory, SWAD, and DG**
> > >
> > > We agree that our proposed algorithm (SWAD) does not directly optimize our theoretical results. However, although the theoretical connection seems slightly weaker than other theoretically motivated methods, our method outperforms all previous methods in DG benchmarks with significant gaps. The empirical observation could be the motivation of the work, but we find the link between the performances and our algorithm in our theoretical results: flatness helps domain generalization. We will revise the paper for a reader to clearly find a link between the domain generalization and the flatness (by the theoretical results in S2), as well as the flatness and the proposed SWAD (by the empirical analyses in S3).
> > >
> > > **Correctness of the technical results**
> > >
> > > First, we do not claim our theoretical results are more novel than [A, B] or other previous domain adaptation methods, but our theoretical results bring the concept of the flatness into DG scenario. We believe our novelty is bridging the gap between two separated communities, DG and flatness.
> > >
> > > Secondly, we do not argue that our theoretical results are universally adoptable (particularly to domain adaptation tasks), but our theoretical results are focusing on the domain generalization scenario as the reviewer pointed out. However, we believe that the restriction to DG scenario does not mean incorrect theoretical results. Please consider that although domain adaptation (DA) and DG look very similar, their base assumptions are very different (DA allows additional domain discrepancy minimization with unseen domain data, while DG does not allow it), so having different theoretical assumptions. Again, we do not argue that our theoretical results are more novel than previous methods, but we are clarifying why the “missing term” happens in the initial review by the reviewer.
> > >
> > > Finally, as we mentioned in the first response, we will clarify all the details we discussed in the revised paper (note that the paper revision is prohibited in the rebuttal period, so the current version does not include the underlying assumptions). We will clearly state the assumptions and the difference between DA theorems and our results in the revised paper. We will upload the detailed revision plan in a few days.

---

> > > > ### Comment · Reviewer_1Nc7 · 2021-09-01
> > > > **Thanks for the response**
> > > >
> > > > I thank the authors for the response and the detailed revision plan.
> > > > If we count the revision plan in, I tend towards acceptance. I have updated my score.

---

### Official Review · Reviewer_aAQf · 2021-07-12

**Rating:** 6
**Confidence:** 4

**Summary:**

This paper studies the domain generalization (DG) problem setting and proposes a modification to the stochastic weight averaging (SWA) method that leads to better performance on DG testbeds. The proposed method, named SWAD, uses the same basic idea as SWA but samples parameters more densely and uses additional techniques for preventing the incorporation of overfit parameters. Some theoretical analysis supports the general thrust of seeking flat minima for generalization. The focus of the paper, however, is empirical, as SWAD demonstrates improved performance compared to many prior methods on several DG testbeds. Furthermore, SWAD can be combined with these prior methods, such as CORAL, for even better performance.

**Limitations And Societal Impact:**

Yes, the authors have adequately addressed limitations and societal impacts in Section 5 and the Appendix.

**Main Review:**

Originality
---
The paper is, perhaps by design, not particularly original. Rather, it makes the observation that an existing set of techniques with a particular goal (seeking flat minima for improved generalization) also has applications to a different goal (DG). This is still, to the best of my knowledge, a novel observation and contribution, and furthermore the modifications made to SWA to arrive at SWAD also appear novel. Nevertheless, these modifications are relatively small, so the method can still be viewed as a version of the general idea of SWA, and thus originality is not the paper's strong suit.


Quality
---
The paper is of relatively high quality, primarily due to the empirical results. I did not carefully check the theory for correctness, but I am unsure as to the significance of the theorems. They appear rather similar to prior theoretical results on how flat minima can lead to better generalization, but with some additional divergence terms thrown in as we are dealing with the DG setting. So the high level conclusion seems to be: standard (in distribution) generalization also helps with DG, so long as the domain divergence is not too large. This is not an interesting result. It would be nice to have stronger theory that more directly connects flat minima to DG, but the current theory does not do this.

Two more minor comments about the theory. First, the reliance on VC dimension is not desirable, since this could be in the billions for the models used and, in particular for DG testbeds, many orders of magnitude larger than the number of training points. Second, I am unsure about the argument in L114-120. "It implies that..." seems to ignore several terms on the RHS of Eq (2) that could be rather large. The final sentence ("Hence, Theorem 2 and...") also seems to require quite a bit more support for such a general statement.

Nevertheless, as mentioned, I am appreciative of the empirical results that the paper provides, including a detailed study on several testbeds and an ablation study that verifies all of the proposed components of the method. I am inclined to view this paper as an empirical contribution, and a fairly nice one that further supports the notion that strong results can be achieved on existing DG testbeds without paying much attention to the domain labels themselves (instead focusing on standard generalization).


Clarity
---
The paper is overall well written and easy to follow. See my comments above for some suggestions regarding the clarity of the theory. My other comments are more minor. First, though the technical description of the overfit-aware sampling strategy in L151-160 is nice, it would be made more clear with additional intuition and descriptive text, e.g., what is each parameter doing specifically and how did this particular instantiation come about? Second, there is a minor typo on L157 (remove the "\mathcal{E}_{val}^{(t_s)} ="). Third, it is not immediately obvious what the significance is of Figure 5.


Significance
---
As noted, I view this paper as a nice (significant) empirical contribution. Perhaps the significance could be further extended by exploring problems beyond DG. Indeed, there seems to be nothing central to the proposed method that ties it to DG, so could it perhaps be applied to other problems? A discussion of this point or, even better, some experiments would further improve the paper.

---
Edit after reading the author response
---
Thanks for your additional comments. My scores remains unchanged and I continue to recommend acceptance.

**Time Spent Reviewing:**

3

---

> ### Author Response · Authors · 2021-08-10
> **Author response for Reviewer aAQf**
>
> We are very grateful for the positive and constructive reviews by Reviewer aAQf, *e.g., our theoretical observation that flatness also helps DG tasks is novel, the paper quality is relatively high, the paper is easy-to-follow, and significant empirical contribution.* We will address all concerns raised by the reviewer and revise the paper accordingly.
>
> ###  Does the theorem simply imply that in-distribution generalization also helps with DG? This is not an interesting result.
>
> We can answer the question by partially yes and no:
>
> - (Partially yes -- in terms of the theorem proof technique) our theoretical contribution is similar to the previous domain generalization literature [A]. Our theoretical results introduce the concept of flatness (i.e., the robust risk minimization) into the results of [A] (i.e., into the DG tasks). We also make a theoretical connection between the empirical robust risk solution (what we are arguing to solve), the empirical risk solution (what previous works have focused on), and the true domain generalization gap (what we are truly interested in). Note that theorems 1 and 2 are not novel by themselves (particularly, compared to [A]). However, our theoretical novelty is introducing the theoretical foundation of bringing the concept of flatness into DG (as the reviewer also agreed).
> - (No -- in terms of empirical results, applying other in-distribution generalization methods does not help DG) As shown in Table 3, the well-known and widely-applicable in-distribution generalization methods, such as Mixup, CutMix, VAT, cannot improve the DG performances, while they consistently show better in-distribution generalization performances. In contrast, as shown in Table 3 and 4, flatness-aware methods (SWA, SAM, SWAD) show consistent performance improvements against the ERM counterpart. These empirical results may imply that a flatter solution is better than a sharper solution, while most of the in-distribution generalization methods cannot improve DG performances in practice. We believe that theoretically showing that RRM solutions are better than ERM solutions in DG tasks will be very interesting future work by itself.
> - (No -- our theoretical results also can be applicable when the domain divergence in the input domain is large) Note that a number of previous DG works have tried to minimize the domain divergence term (the second term in the right-hand side of Theorem 1) in the feature-level by applying domain agnostic feature learning, e.g., DANN [9] or CORAL [36]. In this case, our theorem is also applicable to the methods reducing the domain divergence [9, 36]. As we discussed in Section 5, we empirically observed in Table 4 that the combination of the flatness-aware solver (SWAD) and the domain divergence reduction method (CORAL) achieved the best performance over all comparison methods.
>
> [A] Ben-David, Shai, et al. "A theory of learning from different domains." Machine learning 79.1 (2010): 151-175.
>
> Overall, although our theoretical proof techniques may not be very novel, our theoretical implication can bring a new research direction to DG tasks; (1) a flat solution help DG, (2) our observation is not against previous theories; considering both a flat solution and reducing domain divergence empirically outperforms all methods. We will revise the discussions in Section 2 accordingly.
>
> ### Answers for the other comments
>
> #### **VC dimension is not desirable**
>
> In practice, we agree with the argument. It could be a common weakness of all theoretical works for deep models based on VC dimension. However, the remaining terms except VC dimension imply that robust risk minimization (i.e., flatness) helps DG. Despite the drawback of VC dimension, we believe that our theoretical results support that flatness helps DG.
>
> #### **L114-120 ignores RHS terms of Theorem 2**
>
> Thanks for the comment. Because other terms in RHS terms of Theorem 2 are constant if the training setting is fixed, the gap between the RRM and ERM is the only factor that controls the upper bound in the inequality. We will revise the main text to better clarity.
>
> #### **SWAD may help other solutions too**
>
> We agree. Especially, SWAD is not model-dependent and does not change the objective function. Since SWAD only affects the learning strategy (high constant learning rate scheduling), we can apply SWAD to general recognition tasks, e.g., ImageNet classification. We did not have enough time to explore SWAD for other tasks yet, but we will try other applications as a future research direction. We will add the related discussion in the main text as the reviewer suggested.
>
> #### **More intuitive description on overfit-aware sampling**
>
> Thanks for the suggestion. We will add more intuitive and graphical description in Figure 2.
>
> #### **Significance of Figure 5**
>
> Figure 5 shows the practical benefit of SWAD; it is free to the model-selection strategy where ERM solutions are very sensitive to the model-selection criterion (as shown in Figure 5). We will revise it to clarify the significance of Figure 5.
>
> #### **Typo**
>
> Thanks for finding the typo, we will revise it.

---

> > ### Comment · Reviewer_aAQf · 2021-08-17
> > **Thanks for your response, it reinforces my original score and recommendation**
> >
> > Thanks for your additional comments. I am still inclined to accepting this paper due to its strong empirical results and simple and clear approach. Based on my understanding, I still do not believe that the theoretical results are particularly insightful. Furthermore, as noted previously, exploring the application of SWAD to other problems could help to increase this paper's significance (and my score), but I do not think it is absolutely critical for acceptance.

---

> > > ### Author Response · Authors · 2021-08-23
> > > **Thank you for the constructive feedbacks**
> > >
> > > We appreciate the reviewer’s constructive comments. We will revise our paper to make the connection between Sections 2 and 3 more clear, and the revision plan will be shared in a few days. As noted by the reviewers, the contribution of the proposed method can be increased through application and analysis to other works. We will do our best to update any additional results possible. We are currently working on ImageNet out-of-distribution robustness benchmarks with SWAD. We will add a comment when we get the results.

---

> > > ### Author Response · Authors · 2021-09-02
> > > **Exploring the application of SWAD to ImageNet robustness**
> > >
> > > Dear Reviewer aAQf,
> > >
> > > We attached additional experiments on ImageNet robustness benchmarks (ImageNet-C, ImageNet-R, and background challenge).
> > > We believe that as the reviewer suggested, SWAD also shows effectiveness on other applications, such as ImageNet tasks.
> > >
> > > We will update our paper to include the results in the discussion section. Thanks for the suggestion.
> > >
> > > -- Authors

---

### Official Review · Reviewer_75KP · 2021-07-17

**Rating:** 6
**Confidence:** 4

**Summary:**

This paper studies how to improve the model performance on domain generalization problems from the perspective of loss landscape. Specifically, the authors propose to apply weight averaging (WA) to find flat minima and demonstrate this could lead to improve model generalization on unseen domains. Theoretically, the authors provide generalization bounds for the proposed method. Empirically, the proposed method achieves better performance on benchmark datasets compared with previous methods.

**Ethical Concerns:**

N.A.

**Limitations And Societal Impact:**

N.A.

**Main Review:**

This paper tries to tackle the domain generalization problem by finding flat minima. It is interesting to investigate the loss landscape of the DNNs on domain generalization problems.

**Originality**: The method proposed in this paper (i.e., SWAD) is mainly built on previous approaches such as SWA, and the authors mention this in the submission.

**Quality**: This submission is a complete piece of work and the claims are well supported by the empirical results. The evaluation is reasonable.

**Clarity**: This submission is well-written and easy to follow.

**Significance**: This paper provides interesting empirical observations and improves the state-of-the-art by applying their proposed new approach.

Pros:
1. The proposed method consistently improves the model performance on domain generalization problems, and the proposed method is flexible and can be easily integrated with other training approaches for better performance.

Cons:
1. The generalization bound does not provide intuition for robust risk minimization (RRM), e.g. the theoretical results do not suggest performing RRM is better than standard ERM.
2. The proposed method is heavily built on the previous approach SWA, which only differs in terms of how frequently the averaging is performed and which iteration is being selected for averaging.

Suggestions:
1. It would be better to put the algorithm (Algorithm 1 in appendix B) in the main body for describing the proposed method.

Missing reference:
1. Reference on generalization bound, [1].

Overall, I think this paper provides a simple approach that can improve model (out-of-domain) performance on domain generalization problems. I would suggest `Marginally above the acceptance threshold'.

[1]. A theory of learning from different domains. Shai Ben-David, John Blitzer, Koby Crammer, Alex Kulesza, Fernando Pereira, Jennifer Wortman Vaughan. Machine learning, 2010.


======================================[updates]======================================

Thanks for the authors' response. I have read the authors' responses as well as the other reviews. I will stick with my score.

**Time Spent Reviewing:**

4

---

> ### Author Response · Authors · 2021-08-10
> **Author response for Reviewer 75KP**
>
> We really thank the positive and constructive feedbacks by Reviewer 75KP, *e.g., the submission is a complete piece of work and the claims are well supported by experiments, the submission is well-written, our empirical observations are interesting, and the performance improvements are significant while our method is flexible and easily integrated with other methods.* We will address all raised concerns by the reviewer and revise the paper accordingly.
>
> ### Intuition of the generalization bound for RRM? (theoretical results do not support that RRM is better than ERM)
>
> In this study, we aim to find the connection between flat solutions (formally formulated by the robust risk minimization) and domain generalization performances. Our theorems 1 and 2 show that although RRM solves a different problem from ERM, the domain generalization gap by the empirical solution of RRM is upper bounded (Theorem 2). From this theoretical observation, we introduce the flatness-aware solvers into DG tasks. As shown in Table 3 and 4, the flatness-aware solvers (SWA [25], SAM [26], and our SWAD) are only methods improving DG performances among generalization methods.
>
> On the other hand, the limitation raised by the reviewer is the drawback of our current theoretical results. We do not guarantee that RRM works better than ERM in theory (but we have empirically shown that a flat solution generalizes better than ERM solutions). Note that, this theoretical limitation (i.e., do not directly compare flat solutions and ERM solutions) is shared with other flatness literatures, such as SAM [26]. However, the purpose of Theorem 1 is showing that although we do not directly solve the ERM problem, RRM also can bound the true risk. From this observation, we introduce a flatness-aware solver into DG tasks.
>
> ### Novelty of the method (SWAD is heavily built upon SWA with minor modifications)
>
> We believe that our contributions are (1) first introducing theoretical results on flatness into the domain generalization setting, and (2) proposing empirically well-performed approximated flatness-aware solver SWAD. Because our purpose is solving DG tasks by flatness-aware solvers, instead of inventing a new method, we first tried a number of flatness-aware solvers, such as SWA [25] or SAM [26]. We also tested SAM [26], but we observed that SWA practically works better than SAM, in terms of both computational cost (SAM requires twice the cost) and empirical performance (64.5% for SAM and 65.5% for SWA).
>
> At the same time, we also observed that directly applying SWA to DG tasks raises two problems; (1) not enough number of gathered weights (2) overfitting. We fix the vanilla SWA to mitigate the issues, and show the empirical benefit by our proposed method. As a result, our SWAD shows +1.4pp improvements compared to the vanilla SWA and +3.6pp improvements compared to the vanilla ERM. As the reviewer pointed out, we do not think our modification is significantly novel (​​we did not focus on developing a perfect flatness-aware solver as discussed in Section 5), but we believe that our technical contribution is making the vanilla SWA practical and stable for DG tasks.
>
> Furthermore, as we mentioned in Section 5, even if one proposes a better flatness-aware solver than SWAD, our contribution is not harmed, because we believe our main contribution is bringing the concept of flatness into DG tasks.
>
> ### Other comments
>
> - We will put the algorithm into the main paper.
> - We apologize for the missing citation; Ben-David (2010). We weren't aware that Ben-David (2010) is missed in the main paper. We will add the classic theoretical reference to the main paper later.

---

> > ### Comment · Reviewer_75KP · 2021-08-19
> > **Quick response**
> >
> > Thanks for the response. After reading the response as well as other reviewer's comments, I still think the theoretical results are not connected to the proposed approach. I do not think the theory results provide new intuitions/understandings on the domain generalization problem. On the other hand, the empirical results in this submission are interesting. Overall, I will stick with my previous score.

---

> > > ### Author Response · Authors · 2021-08-23
> > > **Thank you for the quick response**
> > >
> > > We appreciate the reviewer mentioning that our empirical results are interesting. We will revise our paper to make the connection between Sections 2 and 3 more clear. We will share our revision plan in a few days.

---

### Official Review · Reviewer_74mu · 2021-07-17

**Rating:** 6
**Confidence:** 3

**Summary:**

This paper provides some theoretical justification that so-called flat solutions exhibit better domain generalization along with a strongly performing training methodology, SWAD, for improving domain generalization. They first prove theorems bounding the domain generalization performance in terms of robust risk minimization. Then they propose an approach to finding flat solutions related to previous work on in distribution generalization, and they thoroughly explore this method with regard to the flatness of solutions found, its domain generalization performance, and also perform ablations of the method.



**Limitations And Societal Impact:**

A thoughtful limitations section was included but societal impacts was not. However, I don’t see particular societal impact concerns specific to this work.

**Main Review:**

The connection between the theoretical work in the first part of the paper and the method developed and tested is fairly tenuous. I appreciate the motivation for finding flatter minima but the paper would be tighter if there were a more direct connection between the flatness term in eq (1) and the algorithm. In particular, does SWAD encourage flatness w.r.t. the first term in eq. (1) more than other measures of flatness? My apologies if I missed this.

How novel/different is Theorem 1 compared to classic results from e.g. Ben-David (2010)? Of course, flatness doesn’t enter into these results but the rest seems somewhat similar.

The performance of the SWAD algorithm seems quite impressive, and I think this makes the paper a solid submission.

**Time Spent Reviewing:**

2

---

> ### Author Response · Authors · 2021-08-10
> **Author response for Reviewer 74mu**
>
> We truly appreciate the positive and thoughtful review by Reviewer 74mu, *e.g., the performance of SWAD is quite impressive*. We will address all raised concerns by the reviewer and revise the paper accordingly.
>
> ### The connection between the theoretical work (RRM) and the method (SWAD)
>
> Our paper consists of two sections; (1) motivating the flat solutions for domain generalization -- S2 (2) improving the existing approximation flatness-aware solver (SWA) by introducing **“dense sampling”** and **“overfit-aware”** scheduling -- S3.
>
> Section 2 introduces the relationship between flatness (robust risk) and domain generalization (DG), which is not yet explored in the community. We invested it by extending well-defined theory [Ben David 2010] with RRM. As a result, we found the target domain generalization gap is bounded by the gap between the RRM and ERM. From the results, we were motivated to seek flat minima for domain generalization.
>
> As far as the authors know, a perfect (i.e., a theoretically guaranteed) flatness-aware solver does not yet exist. A number of approximated solvers have been proposed, e.g., SWA [25], SAM [26], and we tried to improve the approximated solvers in this study. Several methods were considered and SWA is chosen to seek flat minima practically. Note that SAM [26] is also tested, which directly minimizes robust risk via approximation. However, we found SWA practically works better, in terms of both computational cost (SAM requires twice the cost) and empirical performance (64.5% for SAM and 65.5% for SWA). We will add the related discussions to the main paper.
>
> Therefore, we decided to use SWA and its improved version, SWAD, achieved 66.9%. SWA(D) is not theoretically guaranteed to reduce robust risk, but the empirical results (Fig 3, and [25]) show it finds flat minima well. We put efforts into describing how we modify the SWA solver in Section 3.
>
> We are also aware that SWAD is not a perfect flatness-aware solver, i.e., it is not a theoretically guaranteed solver, as discussed in Section 5. Please note that we do not argue that SWAD guarantees to reduce robust risk (the first term in eq (1)). Instead, we have shown that SWAD brings huge empirical benefits on flatness (Fig 3), and domain generalization (Table 1-5). However, even if a perfect flatness-aware solver is proposed, our theoretical contribution still holds: showing the relationship between flat minima and DG (Theorem 1 and 2).
>
> We believe that our contributions are (1) first introducing theoretical results on flatness into the domain generalization setting, and (2) proposing empirically well-performed approximated flatness-aware solver SWAD.
>
> ### Difference between Theorem 1 and Ben-David (2010)
>
> First, we apologize for the missing citation; Ben-David (2010). We weren't aware that Ben-David (2010) is missed in the main paper. We will add the classic theoretical reference to the main paper later.
>
> The novelty of Theorem 1 and 2 are as follows: (1) we introduce the robust risk minimization (or flatness) into domain generalization tasks (2) we make a connection between the empirical robust risk solution (what we are arguing to solve), the empirical risk solution (what previous works have focused on), and the true domain generalization gap (what we are truly interested in). Note that theorems 1 and 2 are not novel by themselves (particularly, compared to Ben-David (2010)). However, as we mentioned above, our theoretical novelty is introducing the theoretical foundation of bringing the concept of flatness into DG. Furthermore, as shown in Tables 3 and 4, flatness-aware methods (SWA, SAM, SWAD) are only generalization methods improving the baseline performances.
>
> Finally, our theoretical results also imply that combining domain gap reduction methods and flatness-aware methods will improve the final DG performances, as shown in Table 4 (CORAL + SWAD), where Ben-David (2010) cannot support the results in Table 3 and 4.

---

### Author Response · Authors · 2021-08-26
**Revision plan**

Dear reviewers and AC, we share our detailed revision plan for our paper.

### Revisions for introduction

- We will clarify our main contributions (empirical results): Our contribution is to introduce the concept of flatness into the domain generalization tasks. As our empirical study showed, other in-domain generalization methods are not effective to domain generalization, but flatness-aware methods are only effective.
- We will clarify that our theoretical results are not particularly novel (that is not our main argument), but the theorems are justifying the introduction of the concept of the flatness into the domain generalization methods. We will add this discussion
- We will clarify how our algorithm is proposed: We do not directly derive our algorithm from the theorem (i.e., the algorithm does not directly optimize the theorem), because optimizing the robust risk minimization is not tractable. Instead, we improved the existing flatness-aware solver SWA where our improvement is focusing on the pitfalls when directly applying SWA to domain generalization tasks (overfitting)

For the revision, we will split the last section of Introduction into three paragraphs. The first paragraph will describe the relationship between flatness and domain generalization tasks. We will clarify that our theoretical results are relying on the previous theoretical results. The second paragraph will describe how our algorithm comes out: there is no perfect flatness-aware solver. We use the previously proposed approximated flatness-aware optimization method (SWA), and enhance its pitfalls when directly applying to DG tasks. We will clarify that we also tested SAM, but SWA shows empirically better performances than SAM in the initial state (this text will go to Section 3). The last paragraph will clarify our contributions again. We will not argue that our theoretical contribution is particularly novel, but our theoretical results show the connection between flatness and domain generalization. Our contribution is mostly focusing on the empirical results: we outperform all previously proposed domain generalization methods in all benchmarks and all evaluation metrics. Furthermore, the combination of our method and the previously proposed method (CORAL + SWAD) even outperforms our results. We hope our empirical findings can make a breakthrough in future domain generalization researches.

### Revisions for theoretical results

- We will add missing citations (theorems from domain adaptation methods) in Section 2, and we will add detailed references in the proof section (as we discussed in the reply to Reviewer 1Nc7)
  - Especially, we will precisely state the difference between the assumptions of previous domain adaptation theorems and our theoretical results. Also, we will clarify that if we consider the same assumption as the previous methods, our theorem will show the similar upper bound as the previous results.
- We will add the label function assumptions in Section 2, and we will discuss the difference between domain adaptation and domain generalization (our scenario)

### Revisions for methodology section

We will clarify how our algorithm is proposed. Our theoretical finding implies that flatness can help domain generalization. We first investigate the existing flatness-aware solvers (SWA, SAM) and their initial results on DG benchmarks. We will describe the pitfalls of existing methods (overfitting) and how we improve SWA by proposing the dense sampling strategy and the overfit-aware scheduling.


### Revisions for additional experiments

As suggested by Reviewer aAQf, we are exploring the application of SWAD to other robustness problems. We have worked on ImageNet classification and its robustness benchmarks. If we have meaningful results, we will add the results in Section 5 (discussion section) with “Exploring the other applications of SWAD”.

---

### Author Response · Authors · 2021-08-30
**Exploring the application of SWAD to other problems**

## Additional experiments

Following the suggestion from reviewer aAQf, we tested our method in ImageNet robustness benchmarks, such as ImageNet-C [A], ImageNet-R [B], and background challenge (BGC) [C].

- **ImageNet-C** measures the robustness against common corruptions such as Gaussian noise, blur, or weather changes. We follow [A] for measuring mean corruption error (mCE). The lower ImageNet-C implies that the model is robust against corruption noises.
- **BGC** evaluates the robustness against background manipulations as well as the adversarial robustness. The BGC dataset has two groups, foreground and background. BGC manipulates images by combining the foregrounds and backgrounds, and measures whether the model predicts a consistent prediction with any manipulated image.
- **ImageNet-R** tests the robustness against different domains. ImageNet-R collects very different domain images of ImageNet, such as art, cartoons, deviantart, graffiti, embroidery, graphics, origami, paintings, patterns, plastic objects, plush objects, sculptures, sketches, tattoos, toys, and video game renditions. Showing better performances in ImageNet-R leads to the same conclusion as other domain generalization benchmarks.


| Model          | ImageNet (Acc) ↑ | ImageNet-C (mCE) ↓ | BGC (Acc) ↑ | ImageNet-R (Acc) ↑ |
| -------------- | :--------: | :----------: | :---------: | :----------: |
| ERM (baseline) |    76.5    |     57.6     |     8.7     |     36.7     |
| SWA            |    76.9    |     56.8     |    10.9     |     37.5     |
| SWAD (ours)    |  **77.0**  |   **55.7**   |  **11.8**   |   **38.8**   |

As shown in the table, our method consistently improves in- and out-of-domain robustness. Compared to the ERM baseline, SWAD improves performances by 0.5pp for in-domain ImageNet validation, 1.9 for ImageNet-C, 2.1pp for ImageNet-R, and 3.1pp for BGC. Compared to SWA, our method shows subtle improvement in the in-domain ImageNet validation set (0.1pp), but it shows significant improvements on the other out-of-domain benchmarks; 0.9 for ImageNet-C, 1.3pp for ImageNet-R, and 0.9pp for BGC. These results are consistent with the results of our paper.

These additional results support that our method is robustly and widely applicable to large-scale applications, such as ImageNet. Furthermore, since SWAD can be easily combined with other methods, the performance can be improved with task-specific methods like SWAD + CORAL in our paper.


### Experiment details

In the experiments, we mostly follow standard ImageNet training recipes. We use ResNet-50 architecture, SGD optimizer with momentum of 0.9, base learning rate of 0.1 with linear scaling rule [D] and polynomial decay, 5 epochs gradual warmup, batch size of 2048, and total epochs of 90. For SWA, the learning rate is decayed to 1/20 until 80% of training (72 epochs), and the cyclic learning rate with 3 epochs cycle length is used for the left 20% of training. SWAD follows the same learning rate decay until 80% of training, but averages every weight from every iteration after 80% of training with constant learning rate.

### References

[A] Hendrycks, Dan, and Thomas Dietterich. "Benchmarking neural network robustness to common corruptions and perturbations." arXiv preprint arXiv:1903.12261 (2019).

[B] Hendrycks, Dan, et al. "The many faces of robustness: A critical analysis of out-of-distribution generalization." arXiv preprint arXiv:2006.16241 (2020).

[C] Xiao, Kai, et al. "Noise or signal: The role of image backgrounds in object recognition." arXiv preprint arXiv:2006.09994 (2020).

[D] Goyal, Priya, et al. "Accurate, large minibatch sgd: Training imagenet in 1 hour." arXiv preprint arXiv:1706.02677 (2017).

---

### Decision · Program_Chairs · 2021-09-27

**Decision:**

Accept (Poster)

**Comment:**

This paper proposes Stochastic Weight Averaging Densely (SWAD) to improve domain generalization. As the name suggests, it is a modification of Stochastic Weight Averaging (SWA) by averaging more densely (every iteration).

Reviewers mostly agree on strength and weaknesses of this work. They all believe that this work is a solid empirical contribution. However, there are two major shortcomings: 1- The novelty is very limited given the minimal changes applied to SWA to get SWAD. 2- The theoretical results seem very unrelated to reasons for the success of SWAD (in particular, as opposed to SWA). I highly recommend rewriting the paper with less emphasize on the theoretical results (maybe moving it to appendix?).

Overall, given the empirical contributions, I'm slightly inclined to accept the paper. I hope authors would take reviewers' suggestion into account to improve the paper.